# Visual CoT: Advancing Multi-Modal Language Models with a Comprehensive Dataset and Benchmark for Chain-of-Thought Reasoning

**Hao Shao**[1,2]   **Shengju Qian**[1]   **Han Xiao**[1]   **Guanglu Song**[2]
**Zhuofan Zong**[1]   **Letian Wang**[3]   **Yu Liu**[2✉]   **Hongsheng Li**[1,4✉]

[1]The Chinese University of Hong Kong   [2]SenseTime Research
[3]University of Toronto   [4]HKGAI under InnoHK

## Abstract

Multi-Modal Large Language Models (MLLMs) have demonstrated impressive performance in various VQA tasks. However, they often lack interpretability and struggle with complex visual inputs, especially when the resolution of the input image is high or when the interested region that could provide key information for answering the question is small. To address these challenges, we collect and introduce the large-scale Visual CoT dataset comprising 438k question-answer pairs, annotated with intermediate bounding boxes highlighting key regions essential for answering the questions. Additionally, about 98k pairs of them are annotated with detailed reasoning steps. Importantly, we propose a multi-turn processing pipeline that dynamically focuses on visual inputs and provides interpretable thoughts. We also introduce the related benchmark to evaluate the MLLMs in scenarios requiring specific local region identification. Extensive experiments demonstrate the effectiveness of our framework and shed light on better inference strategies. The Visual CoT dataset, benchmark, and pre-trained models are available on this webpage to support further research in this area.

## 1 Introduction

With the success of large language models (LLMs) like GPT-4 [1] and Gemini [63], researchers are enhancing these models by incorporating visual understanding capabilities. This enthusiasm has led to the emergence of multi-modal large language models (MLLM), such as LLaVA [39, 40], SPHINX [17, 37], and Qwen-VL [3]. Involving the extraction of visual tokens from input images, these MLLMs mostly follow a two-stage schedule: first the alignment of these tokens with linguistic modalities, and then the joint processing in LLMs. MLLMs have demonstrated viability in various scenarios, such as image captioning, visual question answering, and optical character recognition, owing to their ability to generate plausible outputs and leverage the extensive knowledge of LLMs.

However, many popular MLLMs [47, 58, 23, 85, 7, 9, 76, 75, 83] and related benchmarks [35, 8, 22, 73, 74] are primarily trained to respond to instructions based on visual inputs, employing a decoder-only autoregressive design as a single black box. While these models exhibit impressive generation capabilities, they suffer from inaccurate information [36] and even hallucinations [18]. Moreover, the black-box design hinders the interpretability of visual-language models. Additionally, the potential of multi-turn in-context capability and the advantages of chain-of-thought [70, 89, 81] for LLMs have not been extensively explored in MLLMs. Some recent works, such as multimodal-CoT [90]

---

✉ Corresponding author.

38th Conference on Neural Information Processing Systems (NeurIPS 2024) Track on Datasets and Benchmarks.

and [80, 79], have shown improvements by incorporating text-level chain-of-thought reasoning or in-context learning. However, it remains uncharted whether existing MLLMs can benefit from chain-of-thought reasoning in the visual understanding process, along with their interpretability remains largely unexplored.

Furthermore, humans comprehend intricate visual information differently, often by focusing on specific image regions or details within a given sample. For instance, when asked for a detailed regional description, humans tend to scan the entire image first, locate the references, and then focus on the targets. In contrast, most MLLMs process aligned image contexts in a fixed-grain manner with a large amount of computation (*e.g.,* CLIP [57], EVA2-CLIP [62], InternVL [12]). To mimic human-like efficient reasoning behaviors, models need to identify image regions containing essential visual details and dynamically zoom in to capture adjusted context, which current MLLMs struggle with, leading them to seek information primarily from the text domain.

Therefore, there is a pressing need to develop methods that can handle multi-turn, dynamic focused visual inputs, while providing more interpretable stages of reasoning to enhance the efficacy and applicability of MLLMs. However, two significant challenges hinder the design of such pipelines: the lack of intermediate visual chain-of-thought supervision in existing visual question-answering (VQA) datasets, and the reliance of popular MLLM pipelines on static image context inputs.

To address these challenges, we develop and release a 438k visual chain-of-thought dataset by annotating each visual question-answer pair with a bounding box. The bounding box highlights the key image region essential for answering the question. We suppose that accurately locating and comprehending this key region will significantly improve MLLM's response accuracy and relevance. Notably, about 98k question-answer pairs include extra detailed reasoning steps. These annotations are designed to instruct the MLLM in a logical, step-by-step process to identify the final bbox and generate the answer. Building on the dataset, we propose a novel pipeline that unleashes the visual CoT reasoning capability of MLLMs, which is designed to identify and output key regions in an image that provides detailed information relevant to the given question. It integrates the understanding of both the original image and detailed local image to generate the final answer. Besides, we provide the corresponding visual CoT benchmark and pre-trained models for reproducibility, aiming to foster further research in the visual chain-of-thought for MLLMs.

To summarize, this paper makes the following contributions:

- We present a visual chain-of-thought dataset comprising 438k data items, each consisting of a question, an answer, and an intermediate bounding box as CoT contexts. Some items also contain detailed reasoning steps. The dataset spans across five distinct domains.

- We propose a novel multi-turn processing pipeline for MLLMs that can dynamically focus on visual inputs and provide intermediate interpretable thoughts.

- We introduce the visual chain-of-thought benchmark for evaluating MLLMs in scenarios where they need to focus on specific local regions or reasons to identify objects.

## 2 Related Works

**Multi-modal LLMs.** Since the advent of large language models (LLMs), their success in various language applications has paved the way for the development of multi-modal large language models (MLLMs), which integrate vision and language modalities. Initially, MLLMs were treated as dispatch schedulers to connect vision expert models, such as VisualChatGPT [71], HuggingGPT [59], and MM-REACT [80], in order to extend language models to other tasks and modalities. More recently, MLLMs have focused on aligning these modalities through extensive training on image-caption pairs or image-question conversations. Notable methods like LLaVA [40] train a projector that maps image tokens to aligned representations of pre-trained LLMs. Other approaches, such as BLIP-2 [32, 31], adopt a query transformer (Q-Former) to learn image embeddings using learnable queries after obtaining image features. MoVA [96] designs an adaptive router to fuse task-specific vision experts with a coarse-to-fine mechanism. In terms of training strategy, recent works [40, 3, 68, 94, 10, 44] commonly employ a 2-stage framework. The first stage involves pre-training on image-caption pairs, while the second stage focuses on alignment by using question-answering triplets. MLLMs have also been extended to various applications, including fine-grained localization [69, 29] such as object detection [86], video understanding [84, 34, 11], and image generation [25, 56].

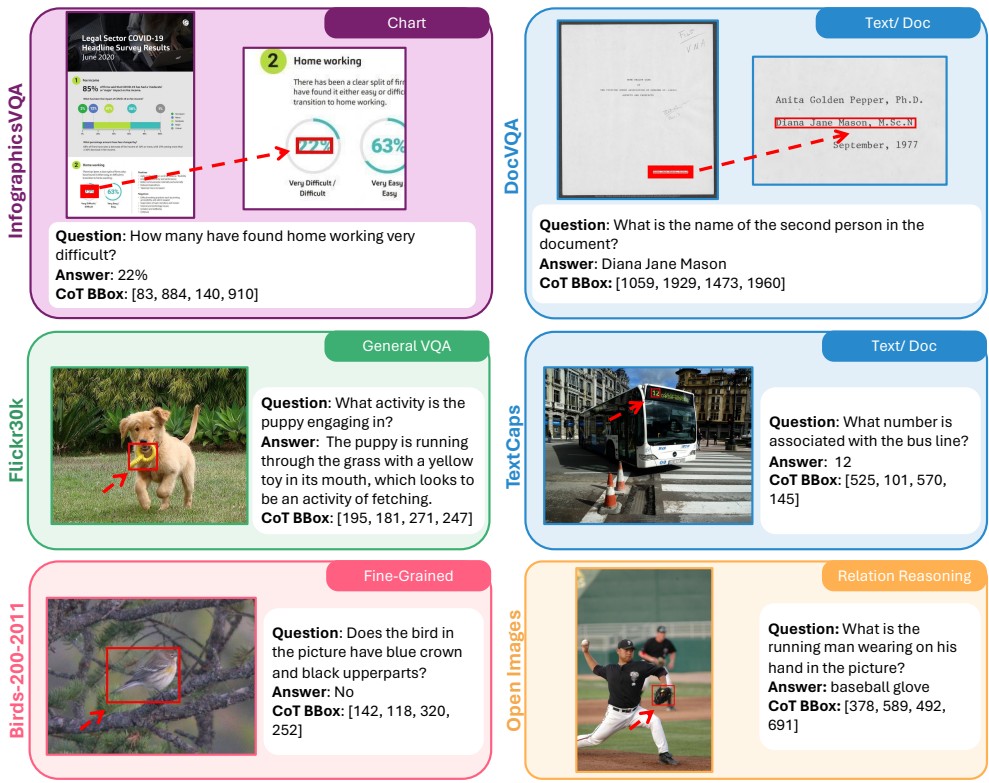

Figure 1: Examples of five domains covered in the visual CoT dataset, with corresponding question-answer annotations and visual CoT bboxes: chart, text/doc, general VQA, fine-grained understanding, and relation reasoning. The red bounding boxes in the images highlight the critical image regions that provide necessary and related information for answering the questions.

**Reasoning Capability of LLMs and MLLMs.** LLMs have demonstrated impressive reasoning capabilities, enabled by in-context learning (ICL) [4], which allows feeding prompted samples and context. This capability has been further enhanced by chain-of-thought (CoT) [70] prompting, which enables LLMs to generate coherent intermediate reasoning steps toward the final answer. Previous studies have shown that LLMs benefit from manually written demonstrations [70] as well as zero-shot prompting outputs [26]. Trar [92] proposes a routing module to dynamically select informative regions based on the attention map. However, due to the domain gap between vision and text data, MLLMs fail to naturally inherit this reasoning capability. To address this limitation, researchers have focused on enhancing the reasoning capability of MLLMs in both the training and prompting paradigms. For instance, Flamingo [2] bridges the gap between these two modalities by pre-training on interleaved visual and textual data. Similarly, other works leverage visual grounded-reasoning [45, 93] data in training, such as Shikra [6] and KOSMOS-2 [53]. More recently, V*[72] and CogCoM[55] modify the general mechanism in MLLMs and collect a series of visual reasoning steps as training data. On the other hand, studies have also explored prompting models [19, 87, 88, 51, 91] to understand complex visual scenes and tasks, focusing on the details of prompting techniques in MLLMs.

## 3 Visual CoT Dataset

There is a shortage of multimodal datasets for training multi-modal large language models (MLLMs) that require to identify specific regions in an image for additional attention to improve response performance. This type of dataset with grounding bbox annotations could possibly help the MLLM output intermediate interpretable attention area and enhance performance. To fill the gap, we curate a visual CoT dataset, as illustrated in Fig. 1 and Tab. 1. This dataset specifically focuses on identifying critical regions within images, a feature essential for models to concentrate on relevant visual elements

Table 1: One data example with detailed reasoning steps, of which we have collected about 98k of this type. The red bounding box shows the important image region for answering the question.

**An example of detailed reasoning steps in GQA dataset**
**Question:** What appliance is to the right of the cabinet?
###
Please think step by step and provide the bounding box coordinate
of the region that can help you answer the question better.
###
**Reasoning steps:** 1. Identify the cabinet in the image.
2. Observe the area to the right of the identified cabinet.
3. Look for any appliance located to the right side of the cabinet.
4. Determine the name of the appliance found in this location
**CoT BBox:** [163, 44, 206, 67]

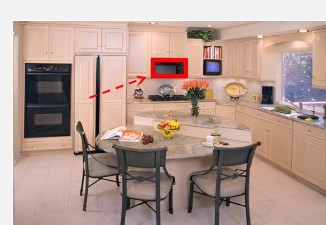

Answer
The appliance is a microwave.

Table 2: The overview of the visual CoT dataset. The dataset spans five distinct domains and includes various source datasets, ensuring a broad representation of visual data styles.

| Domain | Source Dataset | Size | Used GPT-4? | Dataset Description |
|---|---|---|---|---|
| **Text/Doc** | TextVQA [61] | 16k | No | Images with text |
| | TextCaps [60] | 32k | Yes | Images with text |
| | DocVQA [50] | 33k | No | Doc Images |
| | DUDE [65] | 15k | No | Doc Images |
| | SROIE [20] | 4k | No | Invoice Images |
| **Fine-Grained Understanding** | Birds-200-2011 [66] | 10k | No | Images of birds |
| **General VQA** | Flickr30k [54] | 136k | Yes | Images |
| | Visual7W [95] | 43k | No | Images |
| **Charts** | InfographicsVQA [49] | 15k | No | Infographic |
| **Relation Reasoning** | VSR [38] | 3k | No | Images |
| | GQA [21] | 88k | Yes | Images **(with detailed reasoning steps)** |
| | Open images [28] | 43k | No | Images |

to improve response accuracy. Each data sample consists of a question, answer, and a corresponding visual bounding box across five domains, as shown in Tab. 2. Some data samples also include extra detailed reasoning steps.

To ensure a robust foundation for detailed visual and textual analysis, our dataset deliberately integrates a diverse selection of data including text/doc, fine-grained understanding, charts, general VQA, and relation reasoning. These data domains are deliberately chosen to cultivate a comprehensive skill set across varied analytical tasks: 1) Text/doc enhances MLLM's capabilities on OCR and contextual understanding, crucial for applications requiring text interpretation in complex environments. 2) Fine-grained understanding aids in identifying and distinguishing subtle differences in visual appearance and patterns. 3) Charts foster the ability to interpret graphical data, which are essential for business and scientific applications. 4) General VQA exposes models to a wide array of visual queries, improving their general usability. 5) Relation reasoning data develops spatial and contextual awareness of MLLMs, vital for interactive and navigational tasks. Together, these modalities ensure the dataset not only fills existing gaps but also enhances the versatility and contextual awareness of MLLMs across varied scenarios.

## 3.1 Data Generation

To collect and build a diverse and comprehensive Visual CoT dataset, we select twelve source datasets across five distinct domains, primarily consisting of Visual Question Answering (VQA) and Image Captioning datasets. We reuse their images and useful annotations, such as question-answer pairs, image captions, and object relations, to aid in building our dataset. The data construction process involves both linguistic and visual annotators to create question-answer pairs, and provide intermediate chain-of-thought bounding boxes indicating the crucial image region for answering the question. For the linguistic annotations, we employ GPT-4 [1], known for its robust language

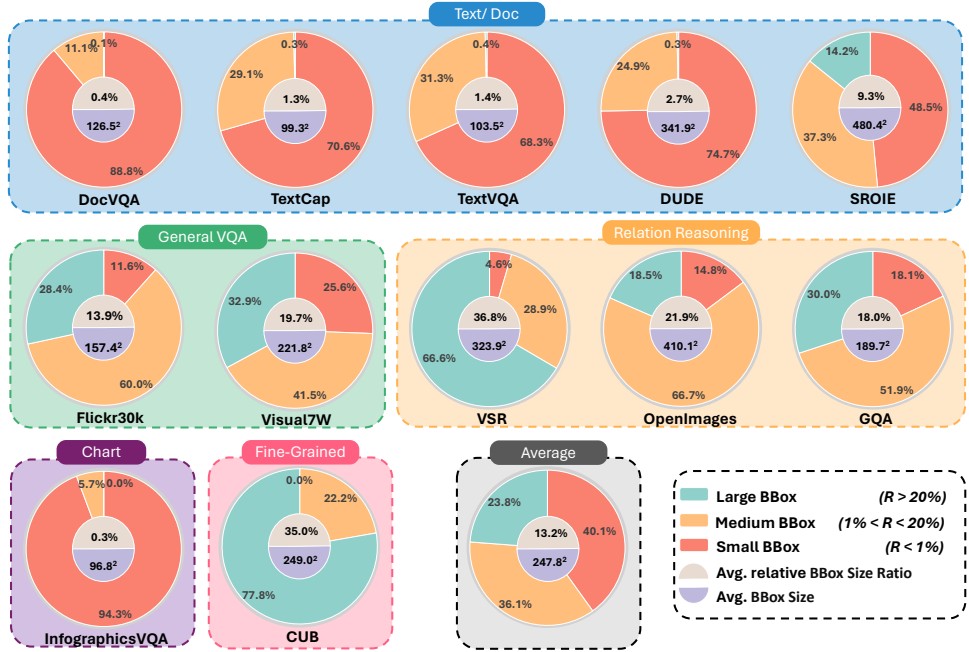

Figure 2: Statistics of the proposed visual CoT dataset. We visualize the CoT bbox distribution, average bbox size, and average relative size of bbox area $R$ for each source dataset.

understanding and generation capabilities. For the visual annotations, we choose PaddleOCR [15], an efficient and accurate tool for optical character recognition. In the following sections, we elaborate on the generation methods employed for each domain-specific dataset.

**Text/Doc.** We choose five text-related datasets to create data in this domain: TextVQA [61], DocVQA [50], DUDE [65], TextCaps [60], SROIE [20]. The five datasets focus on text recognition and comprehension in a variety of images and documents. TextVQA, DocVQA, DUDE and SROIE have already provided question-answer pairs, which we directly adopt. TextCaps, providing only captions and OCR tokens, required us to employ a linguistic annotator to create corresponding questions and answers (see further details in Appendix E.1). For the visual CoT bboxes, we then apply PaddleOCR[15] to detect OCR-identified regions in the image, and specify the CoT bounding boxes as the region that consists of words and sentences aligning with the answer. Furthermore, we also design a filtering pipeline to improve content quality. This process ensures that the areas highlighted by the bounding boxes are directly relevant to the questions.

**Fine-Grained Understanding.** For this domain, we use Birds-200-2011 [66], which is a widely-used dataset for fine-grained visual categorization. This dataset is not only rich in visual data but also includes detailed annotations about various bird parts and their attributes, along with bird bounding boxes in each picture. To leverage this dataset for our MLLM, we have formulated questions that challenge the model to identify specific characteristics or features present in the birds. These questions are designed to test the MLLM's ability to discern and recognize fine-grained details in the images.

**General VQA.** We use Flickr30k [54] and Visual7W [95] as the dataset for general VQA tasks. In Flickr30k, each image encompassed five captions and the bounding boxes of most objects mentioned in the captions. Employing a similar approach to TextCaps, we use GPT-4 to generate questions that require focusing on small objects in the images. The visual CoT bounding boxes in our proposed dataset correspond to the bboxes of objects identified and annotated in the official dataset. Visual7W has already provided the question-answer pairs with object-level grounding annotations.

**Charts.** We select the InfographicsVQA [49] dataset for its high-resolution infographics, which are advantageous for training MLLMs to pinpoint answer locations. Like in our Text/Doc data, we apply OCR techniques to identify regions containing the answers, using these identified areas as the CoT bounding boxes for more precise model training.

**Relation Reasoning.** We select the Visual Spatial Reasoning (VSR) [38], GQA [21], and Open Images [28] datasets to construct data focusing on relation-reasoning. These datasets are rich in spatial relational information among objects in images. For our chain-of-thought (CoT) bounding boxes, we use the bounding boxes surrounding the objects relevant to the question. For instance, if the question is *"What is the material of the desk left to the woman?"*, the bounding box of the desk to the woman's left is designated as the visual CoT bounding box, providing more visual context for the MLLM's reasoning process. In GQA [21] each image is associated with a scene graph of objects and relations. Each question comes with a structured representation of its semantics. With these annotations, we utilize GPT-4 to generate detailed reasoning steps, as illustrated in Tab. 1. The related prompt is available in Appendix E.3.

## 3.2 Dataset Analysis

We provide a visualization of the data statistics in Fig. 2. We partition the bboxes in each dataset into three groups (large, medium, small) based on the relative bounding box size $R$, which is the ratio of the CoT bbox size relative to the total image size. The visualization reveals that the majority of the annotated key regions, particularly in text-oriented datasets, occupy only a small portion of the entire image, highlighting the importance of identifying these crucial areas to enhance performance. Specifically, the average bounding box size is $247.8^2$ pixels, which well aligns with the common input resolution for a vision encoder ranges between 224 and 336 pixels, while the original image size is usually too large and needs down-sampling that loses information. These regions account for only about 13.2% of the image area. This highlights the necessity for MLLMs to accurately pinpoint these crucial areas to enhance processing efficiency and effectiveness. If the model fails to correctly identify and focus on these key regions, the majority of the image processed could be irrelevant, leading to inefficient computation, hallucination, and potential degradation in performance.

## 4 Enhancing MLLMs with Chain-of-Thought Capabilities

Along with the visual CoT dataset, we also propose a visual CoT MLLM framework named VisCoT, which employs standard models without specialized modifications, serving as a baseline to enhance MLLMs with visual CoT capabilities. In this section, we briefly introduce the framework, and illustrate the pipeline in Fig. 3. Readers are referred to Appendix B for more details.

**VisCoT Pipeline.** To train the MLLM baseline with visual CoT data, we add a CoT prompt (*"Please provide the bounding box coordinate of the region that can help you answer the question better."*) to the question, asking the model to identify the most informative region of the image. VisCoT then determines this region and generates its bounding box. During the training phase, we utilize the ground truth bounding box to extract visual information rather than a predicted one in the following steps. With the original image $X_0$ and the bbox, a visual sampler extracts the localized image $X_1$ containing detailed information. The same vision encoder and projector are then used to extract visual tokens $H_1$. The MLLM then integrates visual tokens from both the original and localized images $\{H_0, H_1\}$ to provide more precise and comprehensive answers. For data without visual CoT annotations, this procedure is omitted as indicated by the dashed box in Fig. 3. Here, the MLLM directly answers based on the input image alone. Our VisCoT baseline is thus adaptable to data in both annotated and non-annotated formats simultaneously.

**Visual Sampler.** Given the original image and the predicted bbox, the visual sampler's role is to accurately select the relevant region that considers the visual encoder requirement and bbox corner cases. We first calculate the center point $[x_0, y_0]$, half-width $w_{half}$, and half-height $h_{half}$ of the bounding box predicted by VisCoT. To capture more context and meet the square receptive field requirement of the CLIP model, $\max\{\max\{w_{half}, h_{half}\}, \text{res}_{half}\}$ is chosen as the sample size $s$. $\text{res}_{half}$ is the half input size of the vision encoder. Consequently, the visual sampler crops the region $[x_0 - s, y_0 - s, x_0 + s, y_0 + s]$ for further processing. During inference, if the calculated cropped box extends beyond the image boundaries, the center point is adjusted towards the center of the image to ensure the box remains within the image frame. This adjustment is important for improving the overall performance, as it can mitigate the impact of any detection inaccuracies.

**Inference.** VisCoT offers two options to generate answers: with or without the visual CoT process. If the CoT feature is not needed, users can simply provide the MLLM with the image and question. To engage the CoT feature, users can append the additional visual CoT prompt after the question.

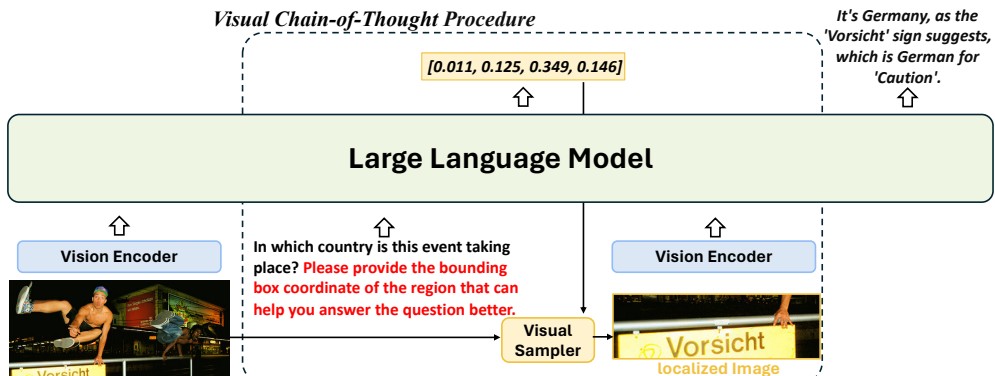

Figure 3: VisCoT first extracts visual tokens from an image and pinpoints the key region relevant to the question. Then, it processes the localized visual information. Finally, the MLLM integrates the information from the overall and localized images to construct a comprehensive and accurate answer.

Table 3: Performance on the Visual CoT benchmark. Datasets highlighted in grey indicate their training splits were not used in our model's training phase. Res indicates input image resolution.

| MLLM | Res. | Doc/Text | | | | | Chart |
|---|---|---|---|---|---|---|---|
| | | DocVQA | TextCaps | TextVQA | DUDE | SROIE | InfographicsVQA |
| LLaVA-1.5-7B [39] | $336^2$ | 0.244 | 0.597 | 0.588 | 0.290 | 0.136 | 0.400 |
| LLaVA-1.5-13B [39] | $336^2$ | 0.268 | 0.615 | 0.617 | 0.287 | 0.164 | **0.426** |
| SPHINX-13B [37] | $224^2$ | 0.198 | 0.551 | 0.532 | 0.000 | 0.071 | 0.352 |
| VisCoT-7B | $224^2$ | 0.355 | 0.610 | 0.719 | 0.279 | 0.341 | 0.356 |
| VisCoT-7B | $336^2$ | **0.476** | **0.675** | **0.775** | **0.386** | **0.470** | 0.324 |

| MLLM | Res. | General VQA | | Relation Reasoning | | | Fine-grained | Average |
|---|---|---|---|---|---|---|---|---|
| | | Flickr30k | Visual7W | GQA | Open images | VSR | Birds-200-2011 | |
| LLaVA-1.5-7B [39] | $336^2$ | 0.581 | 0.575 | 0.534 | 0.412 | 0.572 | 0.530 | 0.454 |
| LLaVA-1.5-13B [39] | $336^2$ | 0.620 | **0.580** | 0.571 | 0.413 | 0.590 | **0.573** | 0.478 |
| SPHINX-13B [37] | $224^2$ | 0.607 | 0.558 | 0.584 | 0.467 | 0.613 | 0.505 | 0.419 |
| VisCoT-7B | $224^2$ | **0.671** | **0.580** | 0.616 | **0.833** | **0.682** | 0.556 | 0.550 |
| VisCoT-7B | $336^2$ | 0.668 | 0.558 | **0.631** | 0.822 | 0.614 | 0.559 | **0.580** |

**Model Training** VisCoT baseline is trained in two stages. In the first stage, consistent with LLaVA-1.5, we freeze the weights of the vision encoder and LLM, and utilize image-text caption data for training. In the second stage, all weights are trainable. For more details, see Appendix B.

# 5 Experiments

Firstly, we provide an overview of the construction and evaluation of the CoT benchmark. Subsequently, in the evaluation phase, we begin by accessing VisCoT on the proposed benchmark (refer to Sec. 5.2). Additionally, we conduct further experiments to analyze the impact of essential components within VisCoT through an ablation study in Sec. 5.3. Finally, we showcase the capabilities of VisCoT in engaging complex multimodal conversations in Sec. 5.4. The training details and detection performance of the visual CoT bboxes can be found in Appendix B & C.

## 5.1 Visual CoT Benchmark

In this section, we provide an overview of our visual CoT benchmark, which primarily focuses on scenarios where the MLLM needs to concentrate on specific regions within a complete image. We utilize 12 source datasets, as shown in Fig. 1, and when an official training/evaluation split exists, we adopt it. In cases where such a split does not exist, we randomly divide the dataset. Additionally, we incorporate the test split of SROIE, DUDE, and Visual7W to evaluate the model's zero-shot visual CoT capabilities. Following the methodology of previous MLLM studies [33, 46], we employ

Table 4: Ablation study on the different BBox selection strategies. 'w/o CoT' indicates a standard, non-CoT-based inference process. 'GT BBox' uses annotated ground truth bboxes. 'Random' and 'Center' refer to using random and center bboxes instead of model predictions.

| BBox Strategy | Doc/Text | | | | | Chart |
|---|---|---|---|---|---|---|
| | DocVQA | TextCaps | TextVQA | DUDE | SROIE | InfographicsVQA |
| Baseline | 0.355 | 0.610 | 0.719 | 0.279 | 0.341 | 0.356 |
| w/o CoT | 0.170 | 0.502 | 0.463 | 0.175 | 0.044 | 0.332 |
| GT BBox | **0.774** | **0.827** | **0.840** | **0.718** | **0.633** | **0.778** |
| Random | 0.208 | 0.463 | 0.495 | 0.157 | 0.146 | 0.378 |
| Center | 0.220 | 0.533 | 0.558 | 0.204 | 0.205 | 0.366 |

| BBox Strategy | General VQA | | Relation Reasoning | | | Fine-grained | Average |
|---|---|---|---|---|---|---|---|
| | Flickr30k | Visual7W | GQA | Open images | VSR | Birds-200-2011 | |
| Baseline | 0.671 | 0.580 | 0.616 | 0.833 | 0.682 | 0.556 | 0.550 |
| w/o CoT | 0.610 | 0.554 | 0.600 | 0.656 | 0.634 | 0.534 | 0.443 |
| GT BBox | **0.692** | **0.699** | **0.796** | **0.896** | **0.792** | 0.577 | **0.752** |
| Random | 0.627 | 0.458 | 0.477 | 0.763 | 0.585 | **0.683** | 0.453 |
| Center | 0.653 | 0.529 | 0.547 | 0.803 | 0.657 | 0.609 | 0.490 |

Table 5: Ablation study on the visual sampler design.

| Expanded Cropping | Centered Cropping | Doc/ Text | Chart | General VQA | Relation Reasoning | Fine-grained | Average |
|---|---|---|---|---|---|---|---|
| | | 0.399 | 0.321 | 0.621 | 0.668 | 0.509 | 0.496 |
| ✓ | | 0.410 | 0.328 | 0.625 | 0.678 | 0.531 | 0.506 |
| | ✓ | 0.434 | 0.331 | 0.641 | 0.677 | 0.521 | 0.518 |
| ✓ | ✓ | **0.461** | **0.356** | **0.626** | **0.710** | **0.556** | **0.550** |

ChatGPT [52] and ask it to assign a numerical score between 0 and 1, where a higher score indicates better prediction accuracy. For detailed information on the prompt used for ChatGPT-based evaluation, please refer to Appendix E.4.

## 5.2    Performance Evaluation

In this section, we comprehensively evaluate VisCoT across various multi-modal tasks to thoroughly assess our model's visual understanding ability. Tab. 3 highlights the enhancements through the visual CoT benchmark. We also showcase the baseline performance of our model on other benchmarks in Appendix D, where it directly answers questions without employing the visual CoT process.

In Tab. 3, we test our model and LLaVA-1.5 on the proposed visual CoT benchmark as detailed in Sec. 5.1. To demonstrate the impact of the chain-of-thought process, we also include the ablation study that removes this reasoning process and directly generates the response in a standard, direct manner. Notably, our pipeline shows significant improvement in the doc/text-related tasks and high-resolution image processing, even when the training splits from corresponding datasets are not utilized for the model training. For instance, SROIE [20] is a dataset that involves extracting key information from scanned receipts, such as the company name and the total price. Our model achieves $8\times$ performance compared to the standard pipeline without a chain-of-thought process. Furthermore, the visual CoT pipeline also shows superior results in other benchmark tasks, showing its efficacy in enhancing the model's comprehensive visual and textual interpretation abilities.

## 5.3    Ablation Study

In the ablation studies below, in default, we ablate VisCoT-7B with a resolution of 224 and mainly evaluate in the proposed visual CoT benchmark.

**Visual CoT BBox Selection Strategies.** Tab. 4 showcases the performance of our model on the visual CoT benchmark using different strategies for bbox selection. As anticipated, employing ground truth annotated bounding boxes instead of model predictions yields the highest performance, surpassing the baseline by a significant margin. This can be considered the upper bound of our model's potential.

**Token Efficiency.** The visual CoT pipeline utilizes double the visual tokens for answer generation, leading us to assess its performance at various resolutions: 224, 336, and 448. As depicted in Fig. 4, the visual CoT pipeline exhibits improved token efficiency in our model. For instance, when equipped with the visual CoT, our model's accuracy at 224 resolution surpasses that of the standard pipeline at 448 resolution, while only using half the visual tokens.

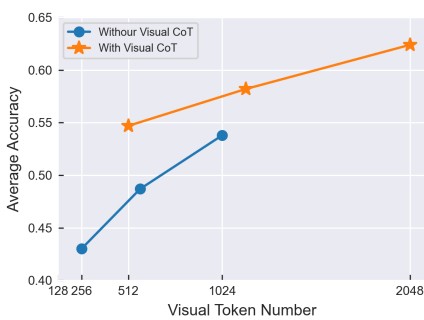

Figure 4: Trade-offs between visual token numbers and average accuracy on the visual CoT benchmark.

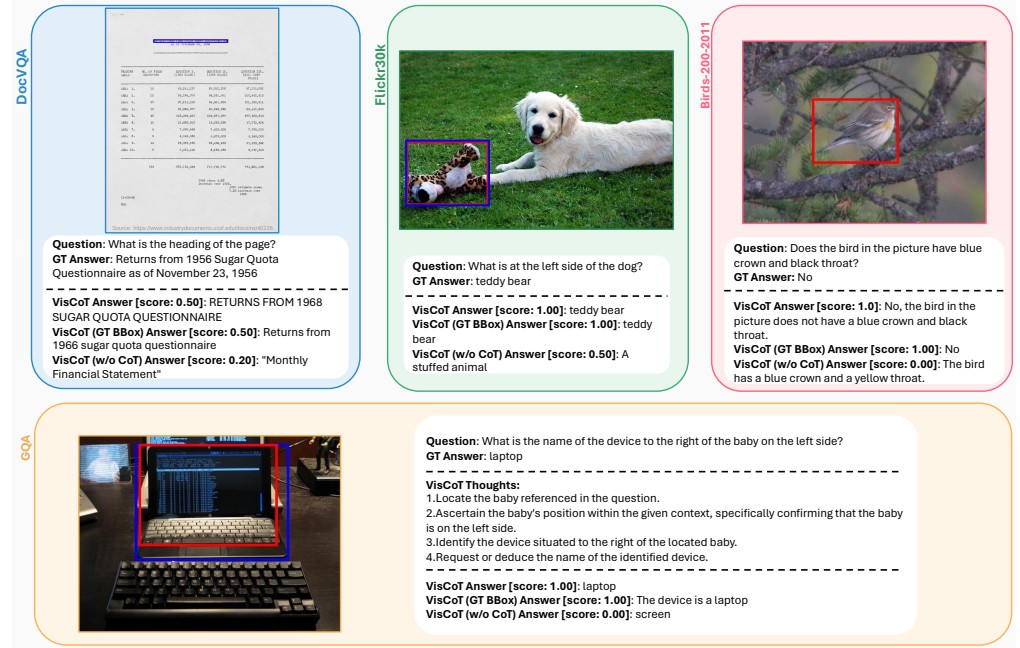

Figure 5: Visualization results of visual CoT to illustrate the difference between various inference modes. Model-generated bounding boxes are shown in red, while ground truth (GT) bounding boxes are in blue. The scores are evaluated by the ChatGPT. Best viewed in color and zoomed in.

Interestingly, random box selection demonstrates similar performance to the 'w/o CoT' approach, suggesting limited impact when the box selection is arbitrary or the prediction is incorrect. However, selecting the 'Center' box exhibits an improvement over the "Random" strategy, indicating that the central region of an image often contains more relevant information. This ablation study provides two key insights: firstly, our model excels at accurately predicting visual bounding boxes, and secondly, the precision of these box predictions significantly influences overall performance.

**Visual Sampler.** We ablate the visual sampler design in Tab. 5. Expanded Cropping refers to enlarging the cropped region if the region is smaller than the vision encoder's input size. Centered Cropping denotes moving the cropped region toward the center if the region extends beyond the image. The results reveal that more image context can bring better performance, and we suppose that it mitigates the problem of detection inaccuracies.

## 5.4 Visualization

This section displays VisCoT's qualitative performance through Fig. 5, highlighting its visual CoT ability to identify critical regions in images that aid in answering questions and synthesizing the combined contexts of both original and zoomed-in images. We also provide comparative results with different configurations: VisCoT (GT BBox), and VisCoT (w/o CoT). The accuracy of detection and depth of understanding directly contribute to the quality of the generated answers.

## 6 Conclusion

In this paper, we introduced VisCoT, a pioneering approach that enhances multi-modal large language models with visual chain-of-thought reasoning. This methodology addresses critical gaps in MLLMs, particularly in interpretability and processing dynamic visual inputs. Our visual CoT dataset offers 438k annotated question-answer pairs for detailed visual analysis. Our novel multi-turn processing pipeline allows MLLMs to dynamically focus and interpret visual data, mirroring human cognition. VisCoT provides more interpretable reasoning stages, and the visual CoT benchmark advances the evaluation of MLLMs' focus on specific image areas. Extensive experiments validate the framework's effectiveness, offering a promising starting point for further exploration in visual CoT.

**Acknowledgement.** This project is funded in part by National Key RD Program of China Project 2022ZD0161100, by the Hong Kong Generative AI Research and Development Center (HKGAI) Ltd under the Innovation and Technology Commission (ITC)'s InnoHK, by General Research Fund of Hong Kong RGC Project 14204021. Hongsheng Li is a PI of HKGAI under the InnoHK.

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

# A Overview

Our supplementary includes the following sections:

- **Section B: Framework details.** Details for model design, implementation and training data.
- **Section C: Detection performance of the visual CoT bboxes.** Details for detection performance for the intermediate visual CoT bounding boxes.
- **Section D: More experiment results.** Additional performance evaluation and performance analysis.
- **Section E: Prompt design.** Prompt for generating the visual CoT dataset and evaluating the performance.
- **Section F: Limitations.** Discussion of limitations of our work.
- **Section G: Potential negative societal impacts.** Discussion of potential negative societal impacts of our work.
- **Section F: More visualization.** More Visualization of our dataset and demos.
- **Section I: Disclaimer.** Disclaimer for the visual CoT dataset and the related model.

Following NeurIPS Dataset and Benchmark track guidelines, we have shared the following artifacts:

| Artifcat | Link | License |
|---|---|---|
| Code Repository | `https://github.com/deepcs233/Visual-CoT` | Apache-2.0 license |
| Data | `https://huggingface.co/datasets/deepcs233/Visual-CoT` | CC BY 4.0 |
| Model Weights | `https://huggingface.co/collections/deepcs233/viscot-65fe883e2a0cdd3c59fc5d63` | Apache-2.0 license |

The authors are committed to ensuring its regular upkeep and updates.

# B Framework details

## B.1 Model details

We choose the pre-trained ViT-L/14 of CLIP [57] as the vision encoder and Vicuna-7/13B [13] as our LLM, which has better instruction following capabilities in language tasks compared to LLaMA [64]. Consider an input original image, we take the vision encoder to obtain the visual feature. Similar to LLaVA [40, 39], we use a simple linear layer to project the image features into the word embedding space to obtain the visual tokens $H_0$ which share the same dimensionality of the LLM.

## B.2 Implementation details

Following the setup described by Vicuna [13], our model undergoes a two-stage training process. In the first stage, we pre-train the model for 1 epoch using a learning rate of 2e-3 and a batch size of 128. For the second stage, we fine-tune the model for 1 epoch on our visual CoT dataset, employing a learning rate of 2e-5 and a batch size of 128. The Adam optimizer with zero weight decay and a cosine learning rate scheduler are utilized. To conserve GPU memory during fine-tuning, we employ FSDP (Full Shard Data Parallel) with ZeRO3-style. All models are trained using $32 \times$ A100s. In the case of training the setting with a 7B LLM and a resolution of 224, the first/second pre-training stage completes within 1/16 hours.

## B.3 Training data details

We train the model on a reorganized Vision-Language dataset. The training data is a composite of three sources: the second stage data from LLaVA, data from Shikra's [6] second stage, and our visual CoT data. The inclusion of data from Shikra, which features various datasets with positional annotations, such as RefCOCO [24] for REC, visual gemone [27] for grounding caption. These datasets can enhance VisCoT's ability to accurately identify and understand locations within images. This enhancement is crucial for tasks requiring precise spatial awareness. We listed all training data in Table 6. We removed the images from the training set that are the same as those in the testing or validation set to prevent potential data leakage. Our training data includes three parts, and they are from LLaVA-1.5, a subset of Shikra, and our proposed visual CoT dataset separately.

Table 6: The overview of our training dataset.

| Dataset | Size | Source Datasets |
|---------|------|-----------------|
| LLaVA-1.5 | 665K | LLaVA, ShareGPT, VQAv2, GQA, OKVQA OCRVQA, A-OKVQA, TextCaps, RefCOCO, VG |
| Shikra | 1.4M | RefCOCO(+/g), VG, PointQA-Local/Twice Visual-7W, Flickr30K |
| Visual CoT dataset | 376K | TextVQA, TextCaps, DocVQA, Birds-200-2011 Flickr30K, InfographicsVQA, VSR, GQA, Open images |

# C Detection performance of the visual CoT bboxes

In Table 7, we present the detection performance based on the predicted CoT bounding boxes. A higher performance indicates that our VisCoT identifies the key regions with greater accuracy.

# D More experiment results

## D.1 Performance evaluation

In Tab. 8 and Tab. 9, we showcase the baseline performance of our model, where it directly answers questions without employing the visual CoT process.

**Multi-modal Large Language Models Benchmarks.** In Tab. 8, we evaluate our model on recently proposed MLLM benchmarks such as MME [16], POPE [36], MMbench [42], ScienceQA [43],

Table 7: Detection performance (Top-1 Accuracy@0.5) on the visual CoT benchmark. The ground truth bounding boxes used for computing the metric are the intermediate CoT bounding boxes annotated in our CoT benchmark.

| MLLM | Res. | Doc/Text | | | | | Chart |
| | | DocVQA | TextCaps | TextVQA | DUDE | SROIE | InfographicsVQA |
|---|---|---|---|---|---|---|---|
| VisCoT-7B | $224^2$ | 13.6 | 41.3 | 46.8 | 5.0 | 15.7 | 7.2 |
| VisCoT-7B | $336^2$ | 20.4 | 46.3 | 57.6 | 9.6 | 18.5 | 10.0 |

| MLLM | Res. | General VQA | | Relation Reasoning | | | Fine-grained | Average |
| | | Flickr30k | Visual7W | GQA | Open images | VSR | Birds-200-2011 | |
|---|---|---|---|---|---|---|---|---|
| VisCoT-7B | $224^2$ | 49.6 | 31.1 | 42.0 | 57.6 | 69.6 | 67.0 | 37.2 |
| VisCoT-7B | $336^2$ | 51.3 | 29.4 | 49.5 | 59.3 | 54.0 | 47.1 | 37.6 |

Table 8: Comparison with SoTA methods on 8 benchmarks. VisCoT achieves the best performance on the most of benchmarks, and ranks second on the other. For a fair comparison, VisCoT generates responses directly, without the visual CoT process. SQA [43]; VQA$^T$: TextVQA [61]; MME$^P$: MME-Preception [16]; MME$^C$: MME-Cognition [16]; POPE [36]; MMB: MMBench [42]; MMB$^{CN}$: MMBench-Chinese [42]. [†] uses 50M in-house instruction-finetuning data. [*] uses multiple vision encoders.

| Method | LLM | Res. | SQA | GQA | VQA$^T$ | POPE | MME$^P$ | MME$^C$ | MMB | MMB$^{CN}$ |
|---|---|---|---|---|---|---|---|---|---|---|
| BLIP-2 [31] | Vicuna-13B | $224^2$ | – | 41.0 | 42.5 | 85.3 | 1293.8 | – | – | – |
| InstructBLIP [14] | Vicuna-7B | $224^2$ | – | 49.2 | 50.1 | – | – | – | 36.0 | 23.7 |
| InstructBLIP [14] | Vicuna-13B | $224^2$ | – | 49.5 | 50.7 | 78.9 | 1212.8 | – | – | – |
| Shikra [6] | Vicuna-13B | $224^2$ | – | – | – | – | – | – | 58.8 | – |
| IDEFICS-9B [30] | LLaMA-7B | $224^2$ | 44.2 | 38.4 | 25.9 | – | – | – | 48.2 | 25.2 |
| IDEFICS-80B [30] | LLaMA-65B | $224^2$ | 68.9 | 45.2 | 30.9 | – | – | – | 54.5 | 38.1 |
| Qwen-VL[†] [3] | Qwen-7B | $448^2$ | 67.1 | 59.3 | **63.8** | – | – | – | 38.2 | 7.4 |
| Qwen-VL-Chat[†] [3] | Qwen-7B | $448^2$ | 68.2 | 57.5 | 61.5 | – | 1487.5 | **360.7** | 60.6 | 56.7 |
| LLaVA1.5 [40] | Vicuna-7B | $336^2$ | 66.8 | 62.0 | 58.2 | 85.9 | 1510.7 | – | 64.3 | 58.3 |
| LLaVA1.5 [40] | Vicuna-13B | $336^2$ | _71.6_ | _63.3_ | 61.3 | 85.9 | _1531.3_ | 295.4 | _67.7_ | **63.6** |
| SPHINX* [3] | LLaMA-13B | $224^2$ | 69.3 | 62.6 | 51.6 | 80.7 | 1476.1 | 310.0 | 66.9 | 56.2 |
| VisCoT | Vicuna-7B | $224^2$ | 68.2 | 63.1 | 55.4 | _86.0_ | 1453.6 | 308.3 | **67.9** | 59.7 |
| VisCoT | Vicuna-13B | $224^2$ | _71.6_ | **64.2** | 57.8 | 85.6 | 1480.0 | 255.4 | 66.9 | 60.5 |
| VisCoT | Vicuna-7B | $336^2$ | 68.3 | 62.0 | 61.0 | **86.5** | 1514.4 | 275.0 | 67.3 | 60.1 |
| VisCoT | Vicuna-13B | $336^2$ | **73.6** | _63.3_ | _62.3_ | 83.3 | **1535.7** | _331.8_ | 67.4 | _61.6_ |

TextVQA [61], GQA [21]. Our model still achieves comparative results across all benchmarks. This performance indicates that the visual CoT data we proposed not only enhances visual comprehension in CoT-specific scenarios but also boosts the model's overall visual understanding in standard inference setups. As demonstrated in Tab. 10, the implementation of visual CoT enables our model to achieve superior performance even with a lower resolution and a reduced number of visual tokens. This finding highlights the efficiency and effectiveness of the visual CoT approach in enhancing model accuracy.

**Visual grounding.** Furthermore, we evaluate VisCoT on REC benchmarks with RefCOCO [24], RefCOCO+ [48], and RefCOCOg [48] datasets. Our model outperforms the previous state-of-the-art models, including the specialist models such as G-DINO-L [41] and UNINEXT [78]. Notably, even with a minimal setup (7B LLM & 224 resolution), our approach outperforms methods that utilize higher resolutions or larger LLM models. This demonstrates that our dataset, enhanced with intermediate bounding boxes, significantly improves the model's precision in locating and understanding referred objects or regions. "Top-1 Accuracy@0.5" refers to the accuracy of a model in predicting the correct bounding box as the top prediction when the Intersection over Union (IoU) between the predicted and ground truth bounding boxes meets or exceeds 50%.

### D.2 Performance analysis

Tab. 4 shows that our baseline with visual CoT performs better than the model without CoT. We further investigate whether different bounding box sizes affect performance improvement. In Fig. 6,

Table 9: Performance (Top-1 Accuracy@0.5) on Referring Expression Comprehension (REC) tasks. For a fair comparison, VisCoT generates responses directly, without the visual CoT process.

| Method | Res. | RefCOCO+ | | | RefCOCO | | | RefCOCOg | |
| --- | --- | --- | --- | --- | --- | --- | --- | --- | --- |
| | | val | test-A | test-B | val | test-A | test-B | val-u | test-u |
| *Specialist models* | | | | | | | | | |
| UNINEXT [78] | $640^2$ | 85.24 | 89.63 | 79.79 | 92.64 | 94.33 | 91.46 | 88.73 | 89.37 |
| G-DINO-L [41] | $384^2$ | 82.75 | 88.95 | 75.92 | 90.56 | 93.19 | 88.24 | 86.13 | 87.02 |
| *Generalist models* | | | | | | | | | |
| VisionLLM-H [69] | - | - | - | - | - | 86.70 | - | - | - |
| OFA-L [67] | $480^2$ | 68.29 | 76.00 | 61.75 | 79.96 | 83.67 | 76.39 | 67.57 | 67.58 |
| Shikra 7B [6] | $224^2$ | 81.60 | 87.36 | 72.12 | 87.01 | 90.61 | 80.24 | 82.27 | 82.19 |
| Shikra 13B [6] | $224^2$ | 82.89 | 87.79 | 74.41 | 87.83 | 91.11 | 81.81 | 82.64 | 83.16 |
| MiniGPT-v2-7B [5] | $448^2$ | 79.97 | 85.12 | 74.45 | 88.69 | 91.65 | 85.33 | 84.44 | 84.66 |
| MiniGPT-v2-7B-Chat [5] | $448^2$ | 79.58 | 85.52 | 73.32 | 88.06 | 91.29 | 84.30 | 84.19 | 84.31 |
| Qwen-VL-7B [3] | $448^2$ | 83.12 | 88.25 | 77.21 | 89.36 | 92.26 | 85.34 | 85.58 | 85.48 |
| Qwen-VL-7B-Chat [3] | $448^2$ | 82.82 | 88.59 | 76.79 | 88.55 | 92.27 | 84.51 | 85.96 | 86.32 |
| Ferret-7B [82] | $336^2$ | 80.78 | 87.38 | 73.14 | 87.49 | 91.35 | 82.45 | 83.93 | 84.76 |
| u-LLaVA-7B [77] | $224^2$ | 72.21 | 76.61 | 66.79 | 80.41 | 82.73 | 77.82 | 74.77 | 75.63 |
| SPHINX-13B [37] | $224^2$ | 82.77 | 87.29 | 76.85 | 89.15 | 91.37 | 85.13 | 84.87 | 83.65 |
| VisCoT-7B | $224^2$ | 85.68 | 91.34 | 80.20 | 90.60 | 93.49 | 86.65 | 85.29 | 86.04 |
| VisCoT-7B | $336^2$ | **87.46** | **92.05** | **81.18** | **91.77** | **94.25** | **87.46** | **88.38** | **88.34** |
| VisCoT-13B | $224^2$ | 86.26 | 91.20 | 80.57 | 91.40 | 93.53 | 87.26 | 86.62 | 86.79 |

Table 10: Performance on VQA benchmarks.

| Model | LLaVA-1.5-7B | VisCoT-7B (w/o COT) | VisCoT-7B | VisCoT-7B (w/o COT) | VisCoT-7B |
| --- | --- | --- | --- | --- | --- |
| Res. | $336^2$ | $224^2$ | $224^2$ | $336^2$ | $336^2$ |
| DocVQA | 21.6 | 14.4 | 39.0 | 29.4 | **49.3** |
| TextVQA | 58.2 | 55.5 | 62.9 | 60.2 | **66.9** |
| ChartQA | 17.7 | 14.2 | 19.2 | 17.5 | **22.8** |

we divide each evaluation dataset into five equal parts based on their relative bounding box sizes. We observe that the visual CoT usually achieve greater improvement when the corresponding bounding box is relative smaller.

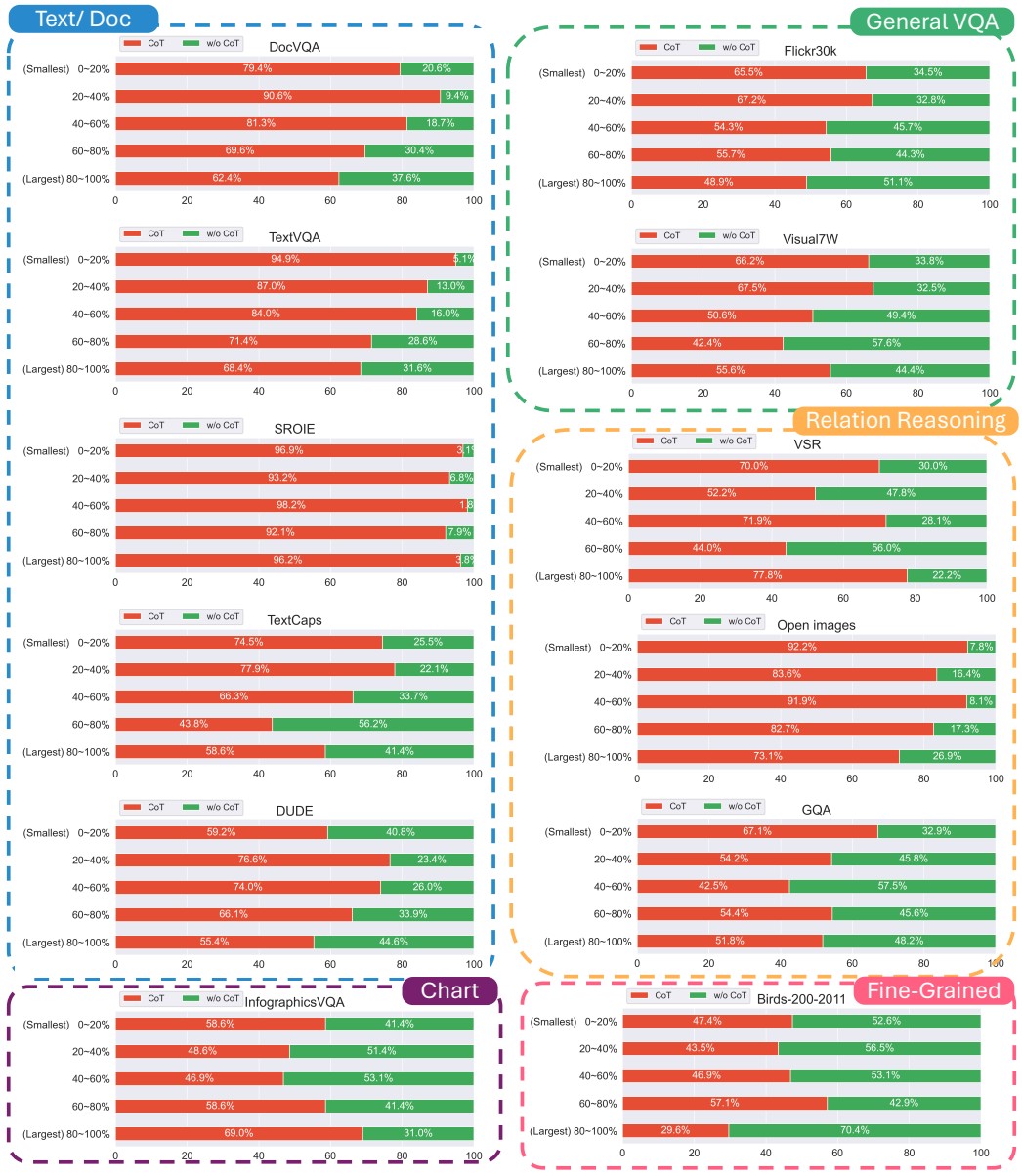

Figure 6: Visualization of performance improvement across different bounding box relative sizes for different source datasets. We find that visual CoT shows a larger improvement in cases where the queried object is relatively small. Red bars represent evaluation data samples where the model with CoT outperforms the model without CoT. Green bars indicate the opposite. The y-axis represents different ranges of relative sizes of bboxes $R$. For example, the 20-40% range indicates that the bboxes in this range occupy the relatively small 20-40% quantile within the entire dataset. For clarity, samples where both models achieve the same scores are omitted.

# E  Prompt design

## E.1  Generating the dataset for TextCaps

You are an AI visual assistant, and you are seeing a single image. What you see is provided with several sentences and Ocr_tokens, describing the same image you are looking at. Ocr_tokens indicates the text in the image. Answer all questions as you are seeing the image. Design a conversation between you and a person asking about this photo. The answers should be in a tone that a visual AI assistant is seeing the image and answering the question. Ask THREE diverse questions and give corresponding answers. Again, do not ask about uncertain details. Do not just makeup questions and answers based on Ocr tokens. Your response should include questions asking about the textual information of the image, the object types, counting the objects, object actions, object locations, relative positions between objects, etc. Please only ask questions that have definite answers:

- One can see the content in the image that the question asks about and can answer confidently;
- One can determine confidently from the image that it is not in the image. Do not ask any questions that cannot be answered confidently.
- One can not see the Ocr_tokens, so the question must not mention 'Ocr'

Craft Questions Around Ocr_tokens: Create questions that directly pertain to these identified words or phrases. Ensure that the question is structured in a way that the answer MUST be a word or phrase directly from the Ocr_tokens. Your answer cannot contain words outside of Ocr_tokens. The answers must be within three words.

Please follow the provided format:
Question: [question]
Answer: [answer]

Here is the context you need to process:
Image description: { }
Ocr_tokens: { }

## E.2 Generating the dataset for Flickr30k

You are an AI visual assistant, and you are seeing a single image. What you see are provided with five sentences, describing the same image you are looking at. Each sentence includes specific objects mentioned and their corresponding locations within the image (*e.g.*, [a peach] is located at [area: 95162] ) Answer all questions as you are seeing the image. Design a conversation between you and a person asking about this photo. The answers should be in a tone that a visual AI assistant is seeing the image and answering the question. Ask diverse questions and give corresponding answers.
The generated questions need closer examination of specific regions in the image to gather detailed information for answering. The generated answers must be based on the corresponding area.
When creating your questions, keep the following considerations in mind:

- Direct Alignment: Ensure the "Focus Area" specified in each question directly corresponds to the content of the question. For instance, if the question refers to "two women", the focus area should align with the portion described as "[Two women]" in the image description.
- Image-Only Basis: Respondents will only have access to the image itself and will NOT see the provided descriptions or area details. Ensure your questions can be answered by viewing the image alone.
- Avoid Repetition: Each question should be distinctive without overlapping content.
- Clarity and Precision: The answers to your questions should be both lucid and exact. Evade vagueness.
- Restricted Question Formats: Refrain from phrasing questions like "What's in region xx?" or "What happens in description 1?". The terms "description" and "region" should not appear in your questions & answers.
- MUST: The "Focus Area" you provide can answer the question you provide.

Please follow the provided format, area_id is a number:
Question: [question]
Focus Area: [area: area_id]
Answer: [answer]

Here is the data you need to process:
Describe 1: With a barn in the background a child puts her head through a hole in a cow cutout and smiles for the camera.
[a barn] is located at [area: 62407]
[a child] is located at [area: 62402]
[a hole] is located at [area: 62405]
. . .

## E.3 Generating the dataset with detailed reasoning steps for GQA

You are an AI visual assistant, and you are seeing a single image. I will provide a question-answer pair along with the corresponding reasoning steps. The question and answer are based on an image. You need to generate the pure reasoning text in a step-by-step format, with each step clearly numbered (1. 2. 3. ... etc). The reasoning text should help solve the question and reach the final answer without including or hinting at the answer itself. The reasoning text must not include any ID numbers.

Here is the data you need to process:

Question: What appliance is to the right of the cabinet?
Answer: The appliance is a microwave.
Reasoning steps: [{"operation": "select", "dependencies": [], "argument": "cabinet (3588933)"}, {"operation": "relate", "dependencies": [0], "argument": "appliance,to the right of,s (1564001)"}, {"operation": "query", "dependencies": [1], "argument": "name"}]

. . .

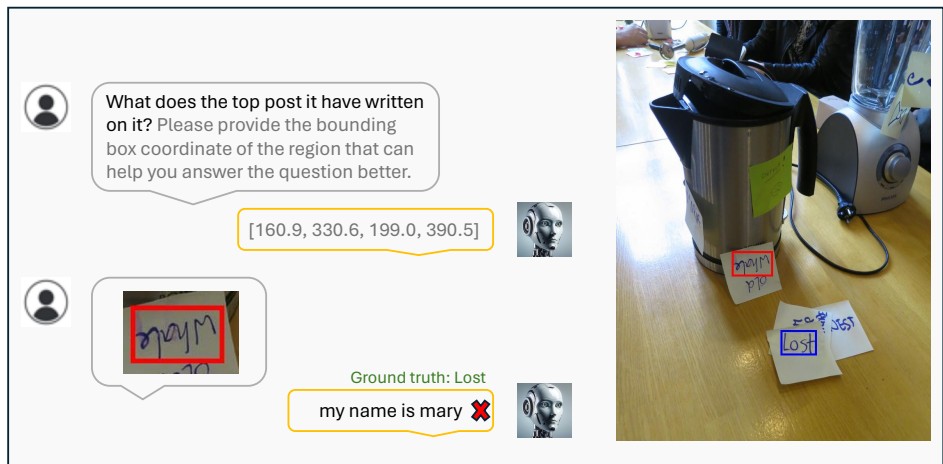

Figure 7: Visualization results of the VisCoT. Model-generated bounding boxes are shown in red, while ground truth (GT) bounding boxes are in blue. In this case, our model incorrectly predicts the CoT region, leading to a wrong answer.

### E.4 Evaluation for the visual CoT benchmark using the ChatGPT

You are responsible for proofreading the answers, you need to give a score to the model's answer by referring to the standard answer, based on the given question. The full score is 1 point and the minimum score is 0 points. Please output the score in the form "score: <score>". The evaluation criteria require that the closer the model's answer is to the standard answer, the higher the score.

Question: { }
Standard answer: { }
Model's answer: { }

## F  Limitations

In scenarios where the input image contains extensive information or the question is particularly complex, VisCoT may struggle to identify the most relevant region for answering the question. As shown in Figure 7, this challenge can sometimes result in the model being misled and producing incorrect responses.

Our data pipeline inherits the limitations of utilizing GPT-4 API. (1) Accuracy and Misinformation: Generated content may not always be accurate, which could lead to the spread of misinformation. To mitigate this, we have designed a comprehensive filtering script as a post-process to improve content quality. (2) Bias and Fairness: Since we do not have access to the training data of GPT-4, the generated instructional data might reflect inherent biases, potentially reinforcing social or cultural inequalities present in the base model training. In terms of data usage, we explicitly state that OpenAI's terms must be adhered to, and the data can only be used for research purposes.

## G  Potential negative societal impacts

The potential negative societal impacts of our work are similar to other MLLMs and LLMs. The development of Visual CoT and MLLMs, while advancing AI, poses societal risks like increased privacy invasion, the perpetuation of biases, the potential for misinformation, job displacement, and ethical concerns regarding accountability and consent.

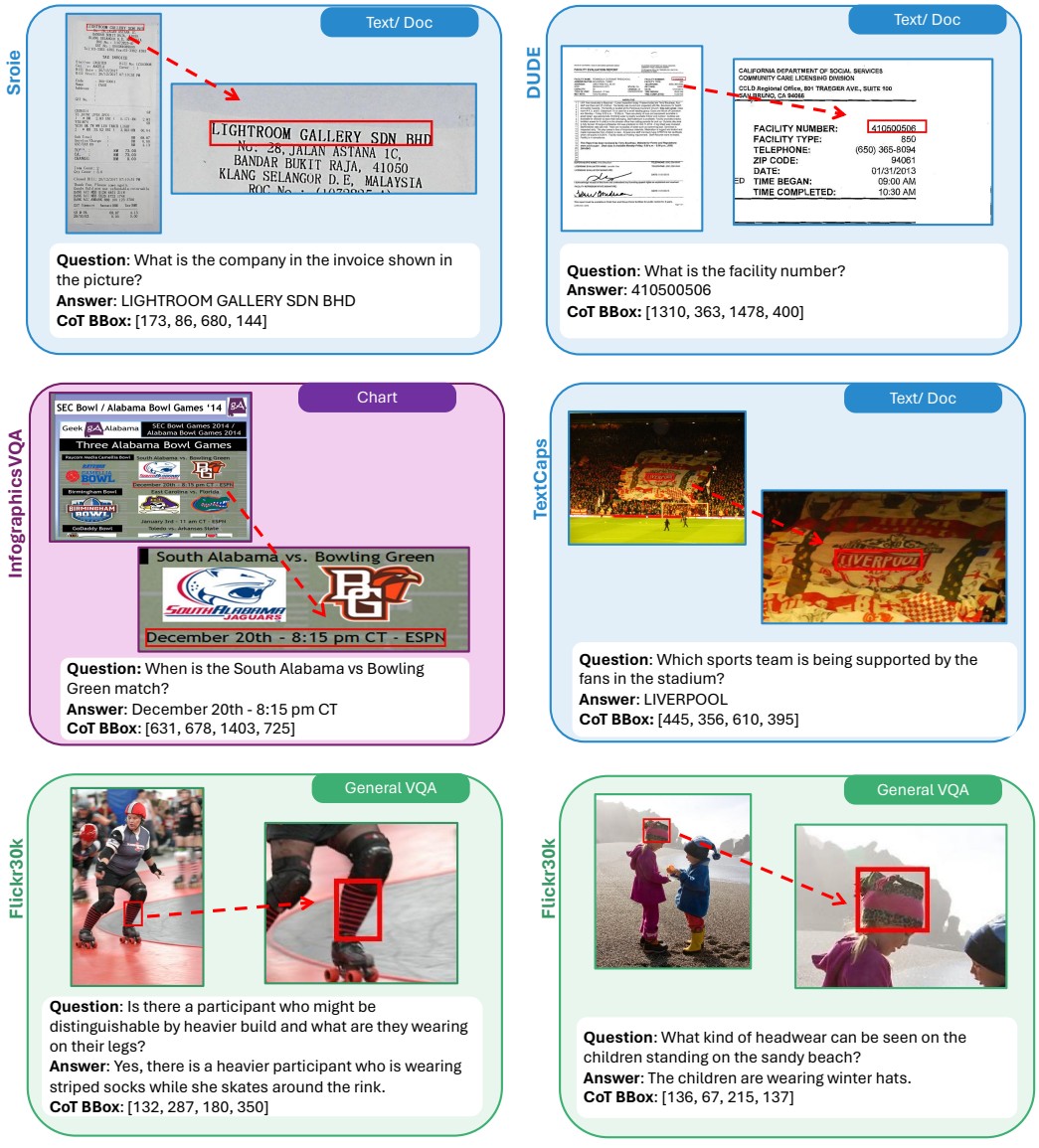

Figure 8: Examples in the visual CoT dataset, with corresponding question-answer annotations and visual CoT bboxes. The red bounding boxes in the images highlight the critical image regions that provide necessary and related information for answering the questions.

## H  More visualization

We provide more visualization results of our proposed visual CoT dataset in Fig. 8, Fig. 9.

We provide more visualization results of our VisCoT baseline in Fig. 10, Fig. 11, Fig. 12, Fig. 13.

## I  Disclaimer

This dataset was collected and released solely for research purposes, with the goal of making the MLLMs dynamically focus on visual inputs and provide intermediate interpretable thoughts. The authors are strongly against any potential harmful use of the data or technology to any party.

**Intended Use.** The data, code, and model checkpoints are intended to be used solely for (I) future research on visual-language processing and (II) reproducibility of the experimental results reported

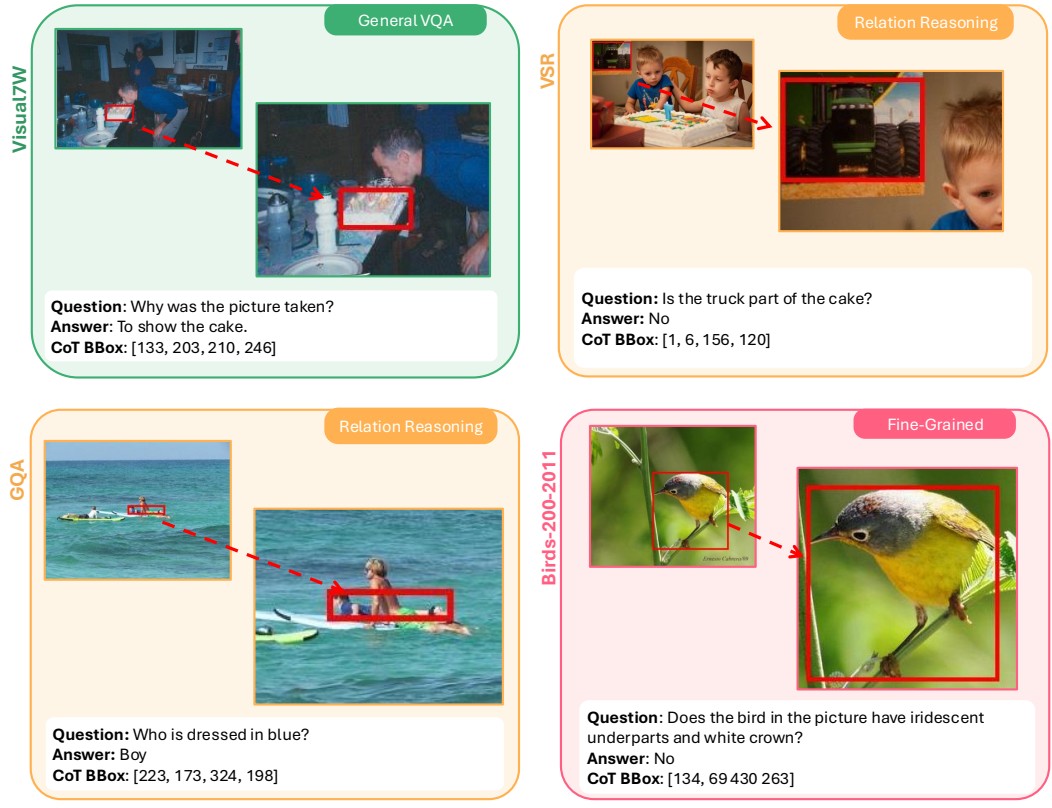

Figure 9: Examples in the visual CoT dataset, with corresponding question-answer annotations and visual CoT bboxes. The red bounding boxes in the images highlight the critical image regions that provide necessary and related information for answering the questions.

in the reference paper. The data, code, and model checkpoints are not intended to be used in clinical care or for any clinical decision making purposes.

**Primary Intended Use.** The primary intended use is to support AI researchers reproducing and building on top of this work. VisCoT and its associated models should be helpful for exploring various vision question answering (VQA) research questions.

**Out-of-Scope Use.** Any deployed use case of the model — commercial or otherwise — is out of scope. Although we evaluated the models using a broad set of publicly-available research benchmarks, the models and evaluations are intended for research use only and not intended for deployed use cases.

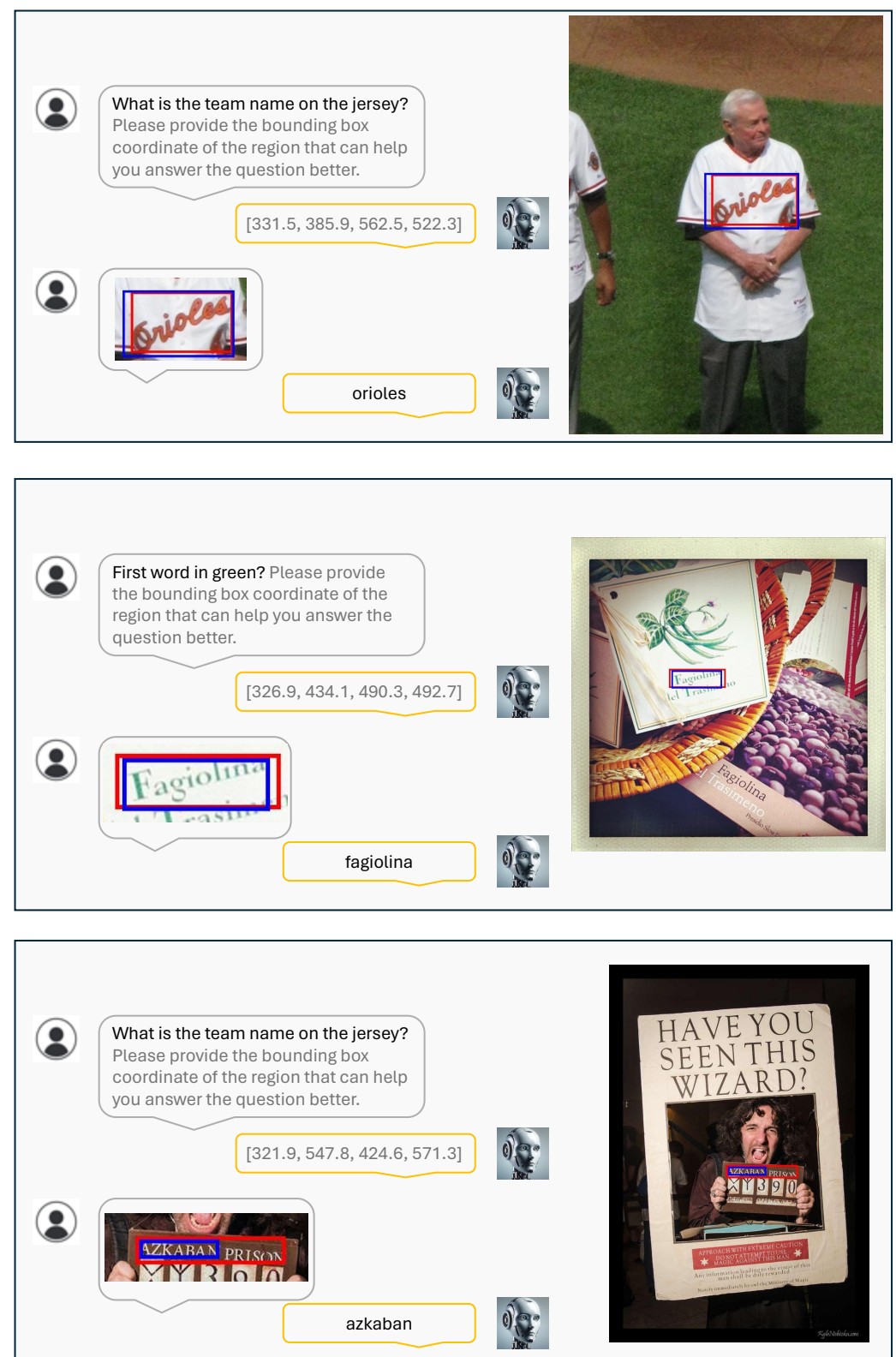

Figure 10: Visualization results of the VisCoT. Model-generated bounding boxes are shown in red, while ground truth (GT) bounding boxes are in blue.

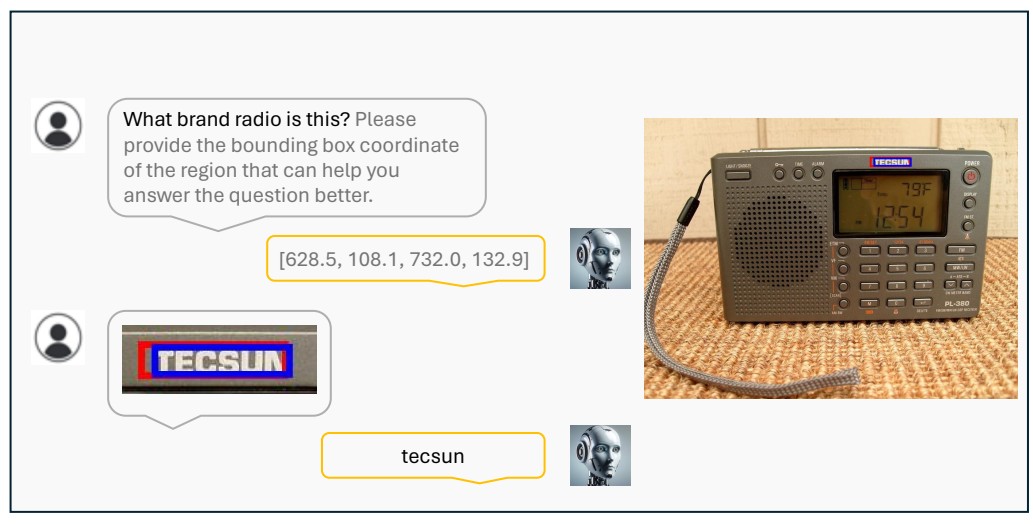

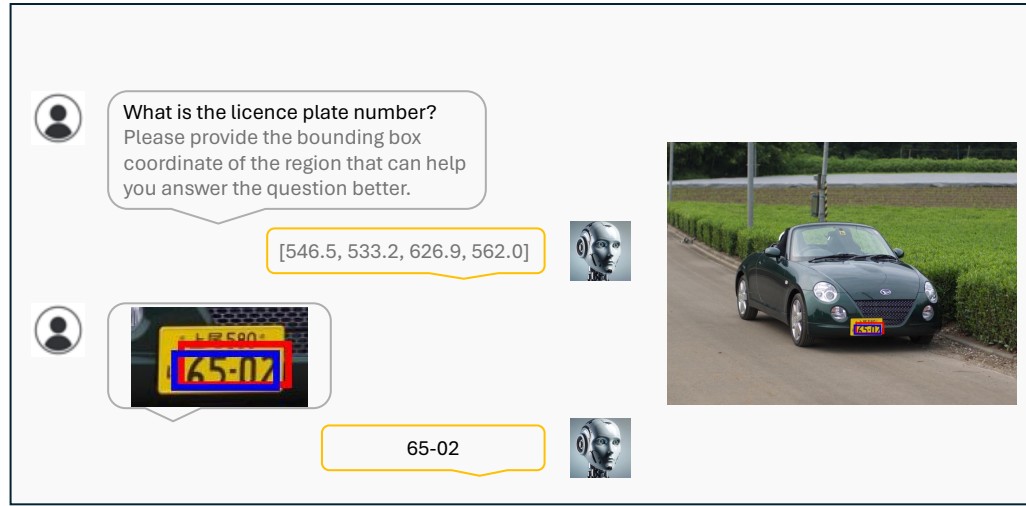

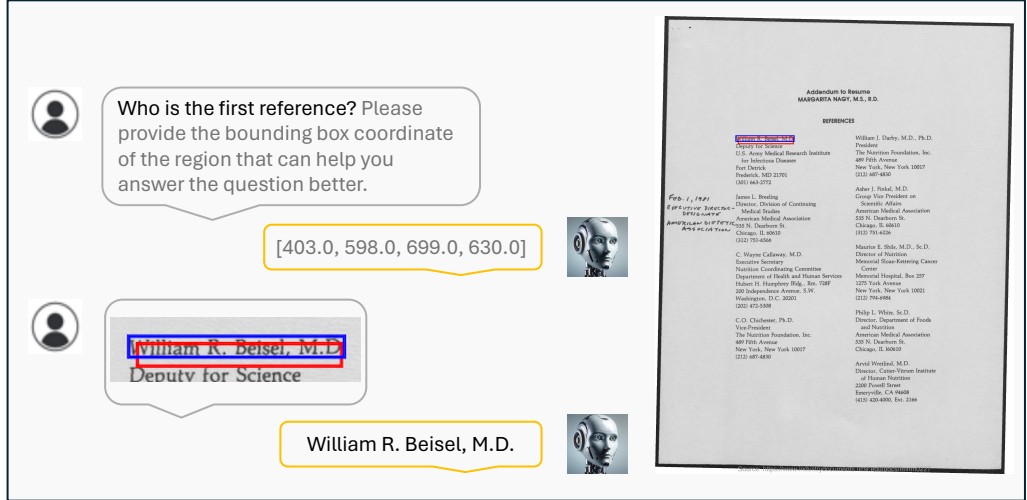

Figure 11: Visualization results of the VisCoT. Model-generated bounding boxes are shown in red, while ground truth (GT) bounding boxes are in blue.

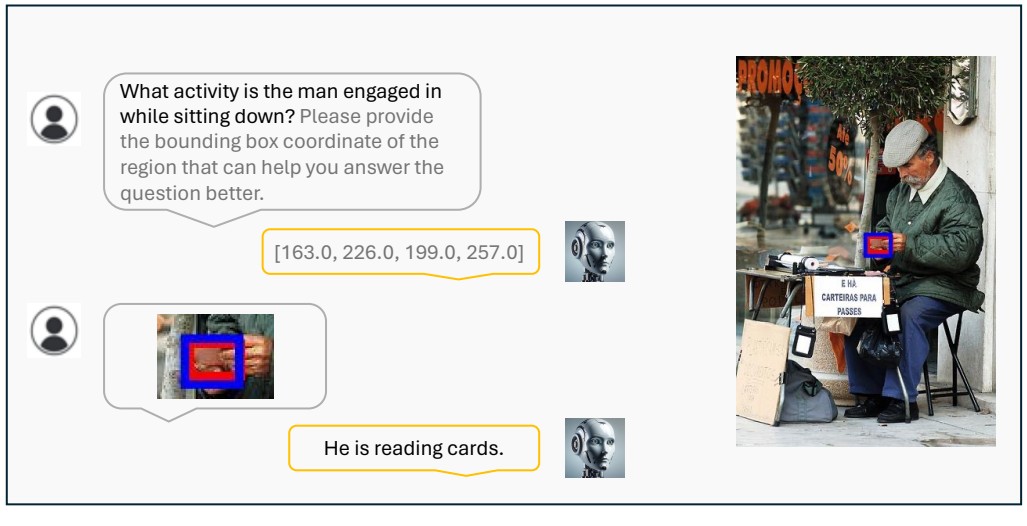

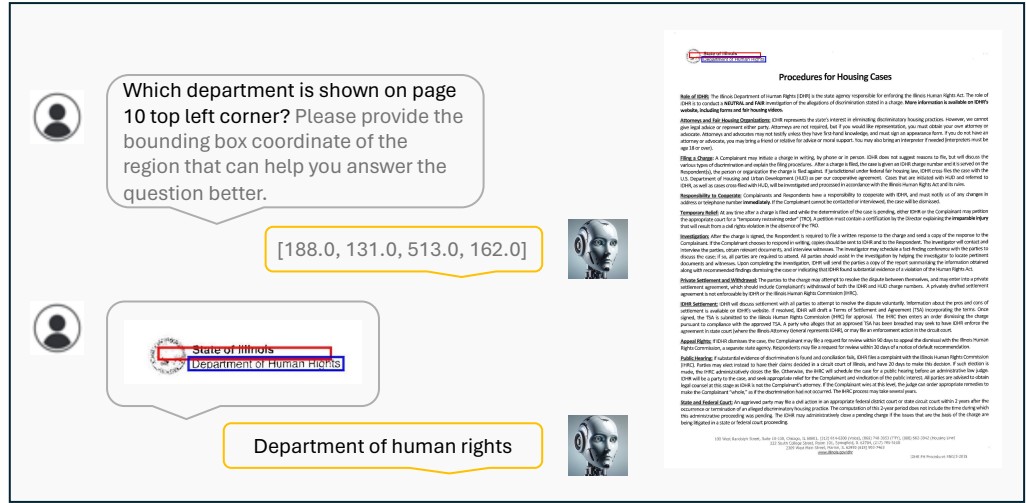

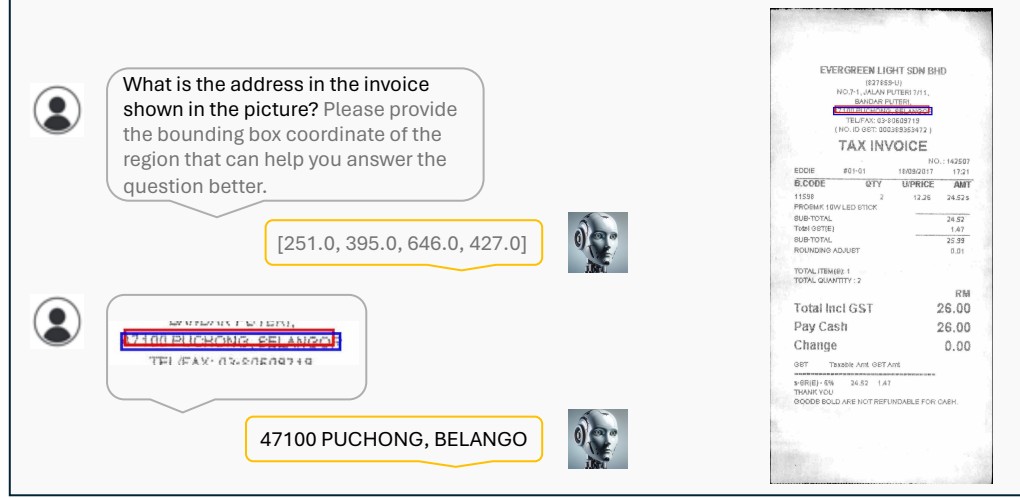

Figure 12: Visualization results of the VisCoT. Model-generated bounding boxes are shown in red, while ground truth (GT) bounding boxes are in blue.

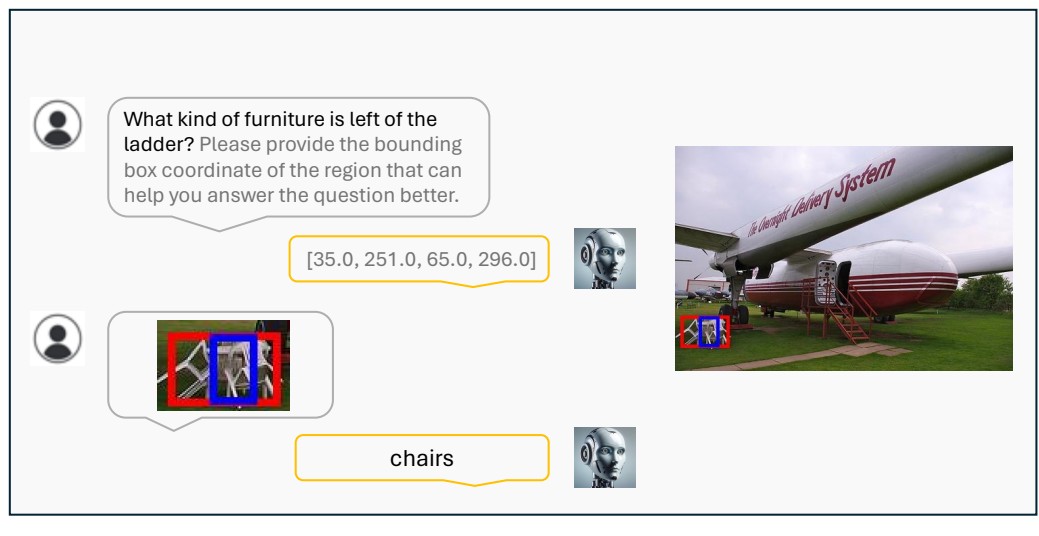

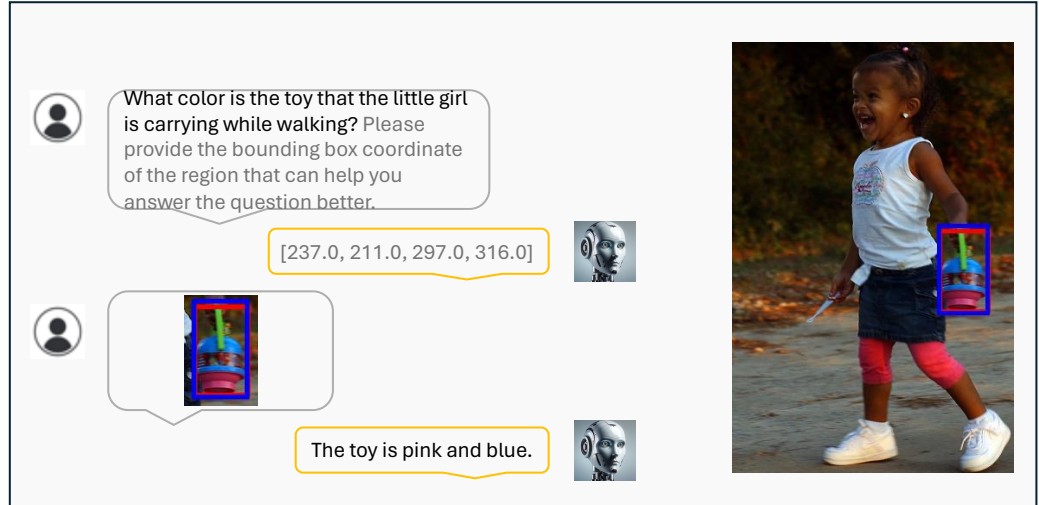

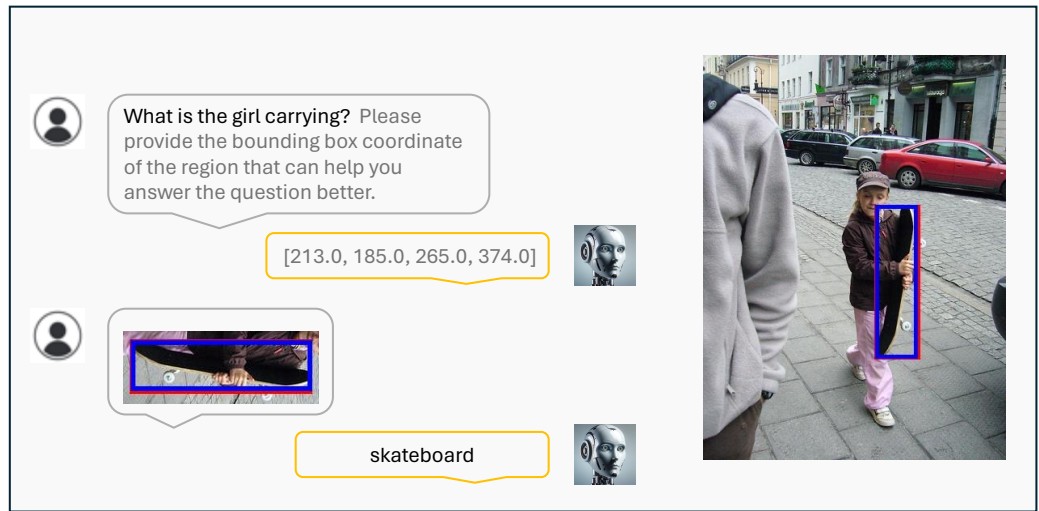

Figure 13: Visualization results of the VisCoT. Model-generated bounding boxes are shown in red, while ground truth (GT) bounding boxes are in blue.

