# OpenReview forum: "Visual CoT: Advancing Multi-Modal Language Models with a Comprehensive Dataset and Benchmark for Chain-of-Thought Reasoning"
_NeurIPS.cc/2024/Datasets_and_Benchmarks_Track — NeurIPS 2024 Track Datasets and Benchmarks Spotlight_

### Official Review · Reviewer_XGWe · 2024-07-24
**Review by Reviewer XGWe**

**Rating:** 6
**Confidence:** 3
**Correctness:** The dataset is constructed in a sound…
**Clarity:** The paper is relatively well organized.

**Review:**

See [Summary And Contributions].

**Strengths:**

1. The paper proposes a large-scale Visual CoT data set. The dataset provides new training and evaluation resources for multimodal large language models (MLLMs), helping to improve the model's ability to interpret and process complex visual inputs.

2. By introducing intermediate reasoning steps and annotations of key areas, the method proposed in the paper can give an explainable reasoning process.

3. The multi-stage processing flow proposed in the paper can dynamically focus on visual input and provide interpretable intermediate thinking results, improving the model's ability to identify specific areas.

**Additional Feedback:**

None.

**Documentation:**

The dataset has relatively complete documentation.

**Ethics:**

No.

**Limitations:**

See [Opportunities For Improvement].

**Opportunities For Improvement:**

1. The method proposed in the paper contains multiple stages, which may increase computational complexity and time overhead. In the case of resource constraints, the practical value of the model may be affected, which should be fully analyzed in the paper.

2. The dataset contains data from multiple fields, but the distribution of data in different fields may be different. It is necessary for the paper to explore the balance of various types of data in the data set.

**Relation To Prior Work:**

The paper describes the differences from previous papers in relatively detail.

**Summary And Contributions:**

The paper proposes a large-scale multi-modal data set and evaluation benchmark called Visual CoT, aiming to improve the performance of multi-modal large-scale language models in chain reasoning. The paper also introduces a novel multi-stage processing framework that allows the model to dynamically focus on key areas of the image and shows significant performance improvements in processing complex visual inputs.

---

> ### Author Rebuttal · Authors · 2024-08-17
>
> We’re grateful for the feedback and suggestions that Reviewer XGWe has made. Here, we provide additional experimental results and explanations addressing reviewer XGWe’s concerns and suggestions.
>
> **Q1: The method proposed in the paper contains multiple stages, which may increase computational complexity and time overhead. In the case of resource constraints, the practical value of the model may be affected, which should be fully analyzed in the paper.**
>
> Thank you for your valuable feedback. We have discussed the token efficiency in Appendix D.1. As discussed, under the same input image resolution, the visual CoT pipeline consists of two stages, resulting in twice the number of visual tokens that the MLLM needs to process. However, our two-stage zoom-in strategy allows the CoT pipeline with lower resolution to outperform the standard pipeline. The first stage in our pipeline focuses on identifying the region that could help the MLLM answer the question, while the second stage uses the global image and the informative local region image to generate the final answer. This approach enables our pipeline to achieve good performance even at low image resolutions.
>
> In Figure 5 of the appendix, we present the trade-offs between visual token numbers and average accuracy on the visual CoT benchmark. To facilitate your review, we have also included the relevant statistics in the accompanying table. The experimental results, derived from 7B models with the same architecture and training dataset but varying input image resolutions, indicate that our pipeline can achieve better performance with an approximate number of visual tokens.
>
> **Standard pipeline**
>
> | Vision tokens | Performance |
> | --- | --- |
> |256 (1 image $ \times 224^2$) | 0.443 |
> |576 (1 image $ \times 336^2$)  | 0.498 |
> |1024 (1 image $ \times 448^2$)  | 0.544 |
>
> **Visual CoT pipeline**
>
> | Vision tokens | Performance |
> | --- | --- |
> |512 (2 images  $ \times 224^2$) | 0.550 |
> |1152 (2 images  $ \times 336^2$)  | 0.580 |
> |2048 (2 images  $ \times 448^2$)  | 0.624 |
>
>
> **Q2: The dataset contains data from multiple fields, but the distribution of data in different fields may be different. It is necessary for the paper to explore the balance of various types of data in the data set.**
>
>
> Thank you for your insightful suggestion. In Figure 2 (main paper), we provide a visualization of the data distribution across various fields, including DocVQA, TextVQA, VSR, GQA, Flickr30k, and others. We analyze the average image size, average relative bounding box ratio, and the distribution of bounding box sizes within each field. Please refer to Section 3.2 for more details.
>
> When designing the Visual CoT dataset, we aimed to incorporate as much data as possible while ensuring a balanced distribution across different fields. Based on your suggestion, we also conducted additional experiments focusing on data balancing during training. The detailed dataset balancing strategies and corresponding experiment results are presented in the following three tables. In addition to the baseline, we performed three ablation studies with different data balancing strategies. These studies were conducted using the 7B VisCoT model with a 224² input image resolution. The baseline refers to the use of the original dataset as described in the paper.
>
> In the first strategy, we balanced the different source datasets as much as possible, especially considering that some source datasets contain fewer than 5k samples. The second strategy focused on balancing the different data domains. Lastly, in the third strategy, we increased the proportion of Text/Doc domain data to investigate whether this would enhance performance within that domain.
>
>
>
> **Data Composition by Source Datasets**
>
> |  Data Balancing strategy   | TextVQA  | TextCaps  | DocVQA  |  DUDE  |  SROIE  | Birds-200-2011  |  Flickr30k  |  Visual7W  |  InfographicsVQA  |  VSR  | GQA  | Open images  |
> |  ----  | ----  |----  | ----  |----  | ----  |----  | ----  |----  | ----  |----  | ----  |----  |
> | Baseline  | 16k | 32k| 33k| 15k| 4k| 10k| 136k| 43k| 15k| 3k| 88k | 43k|
> | Strategy 1  | 16k | 22k| 23k| 15k| 4k| 10k| 46k| 24k| 15k| 3k| 33k | 24k|
> | Strategy 2  | 10k | 18k| 19k| 9k| 4k| 10k| 45k| 15k| 15k| 3k| 37k | 20k|
> | Strategy 3  | 16k | 32k| 33k| 15k| 4k| 10k| 60k| 20k| 15k| 3k| 44k | 22k|
>
>
> **Data Composition by Domains**
>
> |  Data Balancing strategy   | Text/Doc  | Fine-Grained Understanding  | General VQA  |  Charts  |  Relation Reasoning  | Total |
> |  ----  | ----  |----  | ----  |----  | ----  | ----  |
> | Baseline  | 100k | 10k| 179k| 15k| 134k| 438k|
> | Strategy 1  | 80k| 10k| 68k| 15k| 60k| 233k| 205k|
> | Strategy 2  | 60k| 10k| 60k| 15k| 60k| 195k|
> | Strategy 3  | 100k| 10k| 80k| 15k| 69k| 274k|
>
> **Performance on the Visual CoT Benchmark**
>
> |  Data Balancing strategy   | Text/Doc  | Fine-Grained Understanding  | General VQA  |  Charts  |  Relation Reasoning  | Average |
> |  ----  | ----  |----  | ----  |----  | ----  |----  |
> | Baseline  | 0.461 | 0.556 | 0.626| 0.356| 0.710| 0.550|
> | Strategy 1  | 0.453 | 0.559|0.608 | 0.368| 0.712| 0.545|
> | Strategy 2  | 0.450| 0.566| 0.609| 0.371| 0.710|0.544|
> | Strategy 3  | 0.467| 0.550| 0.612| 0.356| 0.705|0.548|
>
> The results indicate that balancing the training dataset across different fields enhances training efficiency. Specifically, Strategies 1 and 2 demonstrate similar performance while utilizing only half the training dataset compared to the baseline. Additionally, Strategy 3, which emphasizes retaining text/doc-related data while reducing the proportion of other fields, shows improved performance.
>
> Overall, both data balancing and data volume significantly influence the model's performance. Due to time constraints, we have only explored these strategies so far. We plan to investigate additional approaches in the future. If you have any further suggestions or concerns, please feel free to share them, and we will do our best to address them.

---

> > ### Author Response · Authors · 2024-08-28
> >
> > Dear Reviewer XGWe,
> >
> > We sincerely appreciate the time and effort you have invested in reviewing our submission. Your insightful feedback has been invaluable to us, and we have diligently worked to address all the concerns you raised in our rebuttal. As the author-reviewer discussion phase is drawing to a close, we would like to confirm whether our responses have effectively addressed your concerns. We are more than happy to provide any further details or explanations. Thank you once again for your thoughtful review and consideration.
> >
> > Best regards,
> >
> > The Authors

---

### Official Review · Reviewer_EkM4 · 2024-07-27
**Visual CoT**

**Rating:** 7
**Confidence:** 4
**Correctness:** Yes
**Clarity:** Yes

**Review:**

Pros:
1. The Visual CoT dataset is a significant contribution, providing a large-scale resource for training and evaluating MLLMs with visual chain-of-thought reasoning.
2. The benchmark is challenging for the existing MLLMs in scenarios requiring specific local region identification and reasoning.
3. The dataset statistics and the in-depth analysis provides the insights of why dynamically focusing on critical visual inputs and interpretable immediate results are important but lacking in most of the MLLMs.
4. The paper is well-written and clear, making it easy to understand the proposed method and its contributions.

Cons:

Please refer to the "Opportunities For Improvement" section.

**Strengths:**

The paper addresses important limitations of existing MLLMs and proposes an effective solution to advance the field of MLLMs. The proposed method and dataset are relevant to a wide range of researchers working on MLLMs, visual reasoning, and interpretable AI. The research is well-conducted, and the proposed method is thoroughly evaluated on a variety of tasks. The paper briefly discusses the potential negative societal impact of the proposed method.

**Additional Feedback:**

None

**Documentation:**

Yes. The paper provides sufficient detail on the data collection process, including the selection of source datasets, the annotation process, and the use of GPT-4 and PaddleOCR. The organization of the dataset into five distinct domains is also well-explained. The appendix provides sufficient detail to support reproducibility, including the release of the code, training data, and checkpoints on GitHub.

**Ethics:**

No concern

**Limitations:**

Yes

**Opportunities For Improvement:**

1. The intermediate CoT consists solely of the coordinates of one bounding box. While it provides a basic level of interpretability by highlighting the relevant image region, it lacks the depth and granularity that a true chain-of-thought process entails. It will be ideal to incorporate more fine-grained annotations that capture the model's reasoning process. This could include:
* Textual explanations: The model could generate brief textual descriptions alongside each bounding box, explaining the reasoning behind selecting that particular region and the visual cues that support the decision.
* Multiple bounding boxes: Instead of a single bounding box, it will be ideal to provide a series of bounding boxes in the reasoning steps when necessary, zooming in on increasingly specific regions of interest. E.g., the bounding boxes of cabinets and potential appliances in Table 1.

2. While Table 4 and Table 5 provide the ablation study of the BBox selection strategies and the visual sampler, it is unclear why moving the center point towards the center of the image when the calculated cropped box extends beyond the image boundaries. What is the likelihood this may happen? Why not directly translate the coordination of the center point based on the diff between the image boundaries and the calculated box?

3. To better justify the need of Visual CoT, it will be ideal to also compare some MLLMs with high resolution, such as Mini-Geimini, and SToA MLLMs, such as GPT-4o, Claude 3.5 Sonnet and Gemini 1.5 Pro.

4. It is unclear what are the metrics being reported in the experiments, and I assume they are accuracy values. It is better to make it clear. Besides, does Top-1 Accuracy@0.5 mean the top-1 predicted BBox has >50% IoU with the groundtruth BBox?

Minor comments:
1. It is better to put the GitHub link to the abstract or in the introduction.
2. The VSR example in Appendix Figure 9 is odd. The question: is the truck part of the cake? but the Answer is "baseball glove".

**Relation To Prior Work:**

Yes

**Summary And Contributions:**

The paper introduces Visual CoT dataset, which includes 438k question-answer pairs annotated with bounding boxes highlighting key image regions and reasoning steps. The proposal of a multi-turn processing pipeline that enables MLLMs to dynamically focus on visual inputs and provide interpretable intermediate thoughts. The benchmark evaluates MLLMs in scenarios requiring specific local region identification and reasoning.

---

> ### Author Rebuttal · Authors · 2024-08-17
>
> We sincerely appreciate the feedback and suggestions provided by Reviewer EkM4. In response, we offer additional experimental results and explanations to address the concerns and suggestions raised by Reviewer EkM4.
>
> **Q1: The intermediate CoT consists solely of the coordinates of one bounding box. While it provides a basic level of interpretability by highlighting the relevant image region, it lacks the depth and granularity that a true chain-of-thought process entails. It will be ideal to incorporate more fine-grained annotations that capture the model's reasoning process.**
>
>
> Thank you very much for your thoughtful suggestions. Regarding textual explanations, we have previously experimented with annotating each bounding box with corresponding textual explanations in the whole dataset. However, it proved very challenging to implement without manual annotation. Nevertheless, with the help of the GQA dataset, we have annotated 100k examples that include textual reasoning processes. We are committed to continuing our efforts to label as many of the remaining 300k examples as possible.
> Concerning multiple bounding boxes, as you mentioned with the example in Table 1, our current dataset includes annotations for each reasoning step, where relevant objects are marked throughout the reasoning process. Your suggestions are highly valuable, and we will continue to explore these possibilities in our future work.
>
>
> **Q2: While Table 4 and Table 5 provide the ablation study of the BBox selection strategies and the visual sampler, it is unclear why moving the center point towards the center of the image when the calculated cropped box extends beyond the image boundaries. What is the likelihood this may happen? Why not directly translate the coordination of the center point based on the diff between the image boundaries and the calculated box?**
>
> 1. Why moving the center point towards the center of the image when the calculated cropped box extends beyond the image boundaries?
>
> I apologize for any confusion caused by our initial description. The purpose of moving the center point inward after the cropped bounding box (bbox) extends beyond the image boundaries is to retain more of the visual information while preserving the content of the original bbox. This adjustment effectively helps to crop more of the central region, which is crucial for preserving the meaningful portions of the image and increasing the method's tolerance for detection error. By making this inward adjustment, we ensure that we capture the important parts of the subject without losing significant details. This approach is particularly useful when the initial bounding box doesn't perfectly align with the image edges or is relatively elongated. In such cases, we first adjust the bounding box to a square shape, which might cause it to extend beyond the image boundaries, making this inward adjustment necessary. We appreciate your understanding and recognize that our initial explanation may not have fully conveyed this intention. We will revise and improve this description in the revised manuscript.
>
> 2. What is the likelihood this may happen?
>
> Thank you for your question regarding the likelihood of this occurring. We have calculated the probability of the bounding box extending beyond the image boundary for each dataset, as shown in the table below. These statistics were obtained using the VisCoT-7B model with a 224² input size as the baseline. The average probability is approximately 39.4%, indicating that the strategy is frequently triggered. We observed that image size significantly impacts this likelihood, with smaller images having a higher probability. Our analysis also shows that image size plays a significant role in this likelihood, with smaller images exhibiting a higher probability. The dataset statistics, which provide more context, are presented in Figure 2 (main paper).
>
> | DocVQA  | TextCaps  |  TextVQA  |DUDE  |  SROIE  | InfographicsVQA  | Flickr30k  |  Visual7W  |   GQA  | Open images  | VSR  | Birds-200-2011  | Average |
> | ----  |----  | ----  |----  | ----  |----  | ----  |----  | ----  |----  | ----  | ----  | ---- |
> | 23.31%  | 25.79%  |  30.23%  |28.86%  |  19.68%  | 40.00%  | 58.99% |  42.31%  |   61.35%  | 39.89%  | 69.31%  | 32.52%  | 39.4% |
>
>
>
>
> 3. Why not directly translate the coordination of the center point based on the diff between the image boundaries and the calculated box?
>
> Thank you for your valuable suggestion.  This is why we have developed the current processing workflow. Initially, we simply cropped the image using the predicted bounding box. However, after a series of experiments—including whether to pad according to the vision encoder’s input size or use the original size—we found that our current strategy significantly improved the model's performance.  Your understanding is absolutely correct, and we greatly appreciate your insightful observation.  We will revise the relevant code and update the paper to make it clearer.

---

> > ### Author Rebuttal · Authors · 2024-08-17
> >
> > # Continue'd...
> >
> > **Q3: To better justify the need of Visual CoT, it will be ideal to also compare some MLLMs with high resolution, such as Mini-Geimini, and SToA MLLMs, such as GPT-4o, Claude 3.5 Sonnet and Gemini 1.5 Pro.**
> >
> >
> > Thank you for your suggestion. We have included a comparison with three open-sourced models: MiniGPT-v2 [1], Qwen-VL-7B [2], and Mini-Gemini [3] and two private MLLMs: GPT-4o, Claude 3.5 Sonnet. The experimental results are presented in the following tables.
> > Our baseline model, Viscot, is configured with 7 billion parameters and an input image resolution of 336². In contrast, the compared models utilize larger image resolutions to capture more fine-grained information. Specifically, MiniGPT-v2 and Qwen-VL-9B both use a resolution of 448, while Mini-Gemini employs two vision encoders with resolutions of 336 and 768, respectively.
> > Qwen-VL-9B features a substantial vision transformer as its vision encoder, consisting of 2 billion parameters. Additionally, its training process leverages large-scale in-house data that is not publicly accessible. The fine-tuning data sample size for Qwen-VL-9B is 50 million, which is 25 times larger than the dataset used in our approach. Notably, 25 million of these samples are OCR data, which significantly enhances the model's performance on text-oriented visual question answering (VQA) tasks.
> >
> >
> > |  Method  | DocVQA  | TextCaps  |  TextVQA  |DUDE  |  SROIE  | InfographicsVQA  | Flickr30k  |  Visual7W  |   GQA  | Open images  | VSR  | Birds-200-2011  | Average |
> > |  ----  | ----  |----  | ----  |----  | ----  |----  | ----  |----  | ----  |----  | ----  | ----  | ---- |
> > | VisCoT-7B (336)  | 0.476 | 0.675 | 0.775 | 0.386| 0.470| 0.324| 0.668| 0.558| 0.631| 0.822| 0.614 | 0.559| 0.580 |
> > |  Open-source MLLMs  | ----  |----  | ----  |----  | ----  |----  | ----  |----  | ----  |----  | ----  | ----  | ---- |
> > | MiniGPT-v2-7B (448) | 0.285 | 0.603 | 0.623 | 0.210 | 0.151 | 0.337| 0.579| 0.583| 0.400| 0.426| 0.554 | 0.539| 0.441 |
> > | Qwen-VL-9B (448) | 0.587  | 0.677   | 0.701   | 0.406 | 0.478 | 0.431 | 0.547 | 0.594| 0.423 | 0.408 | 0.551 | 0.612| 0.537|
> > | Mini-Gemini-7B (336+768)  | 0.314 | 0.614 | 0.627 | 0.310| 0.156| 0.404| 0.627| 0.594| 0.552| 0.454| 0.628 | 0.548| 0.486 |
> > |  Proprietary MLLMs  | ----  |----  | ----  |----  | ----  |----  | ----  |----  | ----  |----  | ----  | ----  | ---- |
> > | GPT-4o  | 0.831 | 0.768 | 0.826 | 0.681| 0.796| 0.751| 0.718| 0.770| 0.801| 0.892| 0.734 | 0.726| 0.774 |
> > | Claude 3.5 Sonnet | 0.846 | 0.806 | 0.831 | 0.663| 0.814| 0.758 | 0.710| 0.784| 0.776| 0.891| 0.720 | 0.757| 0.780 |
> >
> > **Q4：It is unclear what are the metrics being reported in the experiments, and I assume they are accuracy values. It is better to make it clear. Besides, does Top-1 Accuracy@0.5 mean the top-1 predicted BBox has >50% IoU with the groundtruth BBox?**
> >
> > We apologize for any confusion regarding the metrics used in our experiments. Below is a summary of the metrics presented in the manuscript:
> >
> > **Tables 3, 4, 5 (main paper)**
> >
> > These tables show the models' performance on the Visual CoT benchmark, with scores ranging from 0 to 1. The scores were generated using ChatGPT, prompted to assign higher scores for better prediction accuracy. We discussed this in Lines 228-230 of the manuscript and will clarify it further in the revised version. For detailed information on the prompts used for ChatGPT-based evaluation, please refer to Appendix E.4.
> >
> > **Tables 7, 9 (supplementary material)**
> >
> > These tables present the detection performance of bounding boxes. Your understanding is correct: "Top-1 Accuracy@0.5" refers to the accuracy of a model in predicting the correct bounding box as the top prediction when the Intersection over Union (IoU) between the predicted and ground truth bounding boxes meets or exceeds 50%.
> >
> > **Tables 8, 10 (supplementary material)**
> >
> > These tables show the performance of models on popular MLLM benchmarks. The metrics are calculated based on the accuracy of the answers. For more details, please refer to the respective papers: SQA [4]; VQA$^{T}$: TextVQA [5]; MMEP: MME-Perception [6]; MMEC: MME-Cognition [6]; POPE [7]; MMB: MMBench [8]; MMBCN: MMBench-Chinese [8]; ChartQA [9]; DocVQA [10].
> >
> > **Q5: It is better to put the GitHub link to the abstract or in the introduction.**
> >
> > Thank you for the suggestion. We have put this link to the abstract in the revised manuscript.
> >
> > **Q6: The VSR example in Appendix Figure 9 is odd. The question: is the truck part of the cake? but the Answer is "baseball glove".**
> >
> > We apologize for the confusion caused by the typo in the answer and have fixed it in the revised manuscript. The correct answer to the question should indeed be "No." The original record in the metafile confirms this, with the following details:
> > {"question": "Is the truck part of the cake?", "answer": "No", "image": "000000273704.jpg", "width": 640, "height": 425, "bboxs": [[1.05, 6.3, 156.36, 119.63]], "dataset": "vsr", "split": "train"}
> >
> >
> > We hope that our response sufficiently addresses your comments. If you have any further questions or concerns, please feel free to reach out to us for further discussion.

---

> > > ### Author Rebuttal · Authors · 2024-08-17
> > >
> > > # Continue'd...
> > >
> > > **Reference**
> > >
> > > [1] ​​Chen, Jun, et al. "Minigpt-v2: large language model as a unified interface for vision-language multi-task learning." arXiv preprint arXiv:2310.09478 (2023).
> > >
> > > [2] Bai, Jinze, et al. "Qwen-vl: A versatile vision-language model for understanding, localization, text reading, and beyond." (2023).
> > >
> > > [3] Li, Yanwei, et al. "Mini-gemini: Mining the potential of multi-modality vision language models." arXiv preprint arXiv:2403.18814 (2024).
> > >
> > > [4] Ji Qi, Ming Ding, Weihan Wang, Yushi Bai, Qingsong Lv, Wenyi Hong, Bin Xu, Lei Hou, Juanzi Li, Yuxiao Dong, et al. Cogcom: Train large vision-language models diving into details through chain of manipulations. arXiv preprint arXiv:2402.04236, 2024.
> > >
> > > [5] Zhengyuan Yang, Linjie Li, Jianfeng Wang, Kevin Lin, Ehsan Azarnasab, Faisal Ahmed, Zicheng Liu, Ce Liu, Michael Zeng, and Lijuan Wang. Mm-react: Prompting chatgpt for multimodal reasoning and action. arXiv preprint arXiv:2303.11381, 2023.
> > >
> > > [6] Zheng Huang, Kai Chen, Jianhua He, Xiang Bai, Dimosthenis Karatzas, Shijian Lu, and CV Jawahar. Icdar2019 competition on scanned receipt ocr and information extraction. In 2019 International Conference on Document Analysis and Recognition (ICDAR), pages 1516–1520.IEEE, 2019.
> > >
> > > [7] Ruipu Luo, Ziwang Zhao, Min Yang, Junwei Dong, Minghui Qiu, Pengcheng Lu, Tao Wang, and Zhongyu Wei. Valley: Video assistant with large language model enhanced ability. arXiv preprint arXiv:2306.07207, 2023.
> > >
> > > [8] Bryan A Plummer, Liwei Wang, Chris M Cervantes, Juan C Caicedo, Julia Hockenmaier, and Svetlana Lazebnik. Flickr30k entities: Collecting region-to-phrase correspondences for richer image-to-sentence models. In Proceedings of the IEEE international conference on computer vision, pages 2641–2649, 2015.
> > >
> > > [9] Ahmed Masry, Xuan Long Do, Jia Qing Tan, Shafiq Joty, and Enamul Hoque. 2022. ChartQA: A Benchmark for Question Answering about Charts with Visual and Logical Reasoning. In Findings of the Association for Computational Linguistics: ACL 2022, pages 2263–2279, Dublin, Ireland. Association for Computational Linguistics.
> > >
> > > [10] Minesh Mathew, Dimosthenis Karatzas, and CV Jawahar. Docvqa: A dataset for vqa on document images. In Proceedings of the IEEE/CVF winter conference on applications of computer vision, pages 2200–2209, 2021.

---

> > > > ### Author Response · Authors · 2024-08-28
> > > >
> > > > Dear Reviewer EkM4,
> > > >
> > > > We sincerely appreciate the time and effort you have invested in reviewing our submission. Your insightful feedback has been invaluable to us, and we have diligently worked to address all the concerns you raised in our rebuttal. As the author-reviewer discussion phase is drawing to a close, we would like to confirm whether our responses have effectively addressed your concerns. We are more than happy to provide any further details or explanations. Thank you once again for your thoughtful review and consideration.
> > > >
> > > > Best regards,
> > > >
> > > > The Authors

---

> ### Comment · Reviewer_EkM4 · 2024-09-01
> **Increase my rating to 7**
>
> Thanks the authors for providing additional results and the clarification!
>
> The experimental findings have substantiated the potential of VisCoT across various model architectures and pre-trained Masked Language Models (MLLMs). I am inclined to elevate my rating to 7. (Regrettably, the review's editing permissions were prematurely closed, depriving me of the opportunity to modify the rating directly within the tool.)
>
> While the limitations pertaining to multiple bounding boxes and the performance gap in comparison to proprietary MLLMs cannot be ignored, they do not diminish the significance of this work. It would be advantageous to incorporate the limitations and comparative analysis discussed in the review into either the primary manuscript or the supplementary material.

---

> > ### Author Response · Authors · 2024-09-01
> > **Appreciation for Your Feedback and Support**
> >
> > Dear Reviewer EkM4,
> >
> > Thank you for your valuable feedback and for raising the rating. We're pleased our additional results highlighted VisCoT's potential across various architectures and MLLMs, along with the performance gap compared to proprietary models. This gap is partly due to the significant differences in model parameters and data scale between our open-source approach and larger proprietary models, which we will continue to explore. We appreciate your insights on the limitations, particularly regarding multiple bounding boxes. We will carefully consider addressing these aspects in the manuscript and supplementary material, and we will also strive to address these issues in future work.
> >
> > Additionally, if it’s convenient for you, the review tool may now allow rating edits.
> >
> > Thanks again for your constructive suggestions and support.

---

### Official Review · Reviewer_1mvY · 2024-07-30
**Review for Submission 557**

**Rating:** 7
**Confidence:** 4
**Correctness:** Yes.
**Clarity:** Yes.

**Review:**

Overall, I found this paper to be well-written, the dataset well-designed and the experiments and ablations were okay. I had a good impression, and hence my **initial rating of 6 (above acceptance threshold)**. That said, it could be stronger in certain aspects (weaknesses), while other aspects were not clear to me (for clarification). Details are below.

**Strengths:**

S1:  Paper is well-written and well-motivated.

S2:  Dataset seems well-designed, and of sufficient size.

S3:  The proposed model outperforms some existing SOTA models.


**Weaknesses:**

W1:  While I try to refrain from bringing up the "too few comparisons" card, in this case it really does seem that at least 1-2 more SOTA models could have been compared to.

W2:  It would have been a stronger work if not just one model were trained, but the same approach done on different base models (e.g. base LLMs), and showed that the improvements still hold true.

**For clarification (may or may not be a weakness ultimately):**

C1:  The Limitations section in the Supplementary Material didn't seem to address this:  What happens if there are multiple bounding boxes needed for a proper reasoning chain, e.g. to answer "What is between [Object A] and [Object B]?"

C2:  Related to C1, how should multiple bounding boxes across different entities (e.g. different pages in a doc) or across different reasoning steps (e.g. "what is above the object to the right of [Object A]") be handled?

**Strengths:**

Pls see review.

**Additional Feedback:**

Nil

**Documentation:**

Based on a brief look at the Supplementary Material, and URLs for code and dataset, documentation seems sufficient.

**Ethics:**

No major concerns.

**Limitations:**

Pls see review.

**Opportunities For Improvement:**

Pls see review.

**Relation To Prior Work:**

Yes.

**Summary And Contributions:**

The paper is a straightforward one, contributing a dataset for visual chain-of-thought reasoning, by provding bounding boxes and reasoning steps. In terms of size, there are 438K annotated examples in the dataset. A model trained on the dataset and using the proposed approach is compared with some existing SOTA like LLaVA and Sphinx. Ablations of the model are performed.

---

> ### Author Rebuttal · Authors · 2024-08-17
>
> We’re grateful for the feedback and suggestions that Reviewer 1mvY has made. Here, we provide additional experimental results and explanations addressing reviewer 1mvY’s concerns and suggestions.
>
> **Q1: While I try to refrain from bringing up the "too few comparisons" card, in this case it really does seem that at least 1-2 more SOTA models could have been compared to.**
>
> Thank you for your suggestion. We have included a comparison with three SOTA models: MiniGPT-v2 [1], Qwen-VL-7B [2], and Mini-Gemini [3]. The experimental results are presented in the following tables.
>
> Our baseline model, Viscot, is configured with 7 billion parameters and an input image resolution of 336². In contrast, the compared models utilize larger image resolutions to capture more fine-grained information. Specifically, MiniGPT-v2 and Qwen-VL-9B both use a resolution of 448, while Mini-Gemini employs two vision encoders with resolutions of 336 and 768, respectively.
>
> Qwen-VL-9B features a large vision transformer as its vision encoder, consisting of 2 billion parameters. Additionally, its training process leverages large-scale in-house data that is not publicly accessible. The fine-tuning data sample size for Qwen-VL-9B is 50 million, which is 25 times larger than the dataset used in our approach. Notably, 25 million of these samples are OCR data, which significantly enhances the model's performance on text-oriented visual question answering (VQA) tasks.
> |  Method  | DocVQA  | TextCaps  |  TextVQA  |DUDE  |  SROIE  | InfographicsVQA  | Flickr30k  |  Visual7W  |   GQA  | Open images  | VSR  | Birds-200-2011  | Average |
> |  ----  | ----  |----  | ----  |----  | ----  |----  | ----  |----  | ----  |----  | ----  | ----  | ---- |
> | VisCoT-7B (336)  | 0.476 | 0.675 | 0.775 | 0.386| 0.470| 0.324| 0.668| 0.558| 0.631| 0.822| 0.614 | 0.559| 0.580 |
> | MiniGPT-v2-7B (448) | 0.285 | 0.603 | 0.623 | 0.210 | 0.151 | 0.337| 0.579| 0.583| 0.400| 0.426| 0.554 | 0.539| 0.441 |
> | Qwen-VL-9B (448) | 0.587  | 0.677   | 0.701   | 0.406 | 0.478 | 0.431 | 0.547 | 0.594| 0.423 | 0.408 | 0.551 | 0.612| 0.537|
> | Mini-Gemini-7B (336+768)  | 0.314 | 0.614 | 0.627 | 0.312| 0.156| 0.404| 0.627| 0.594| 0.552| 0.454| 0.628 | 0.548| 0.486 |
>
>
> **Q2: Could the approach be further validated by training on different base models (e.g., base LLMs) to demonstrate that the improvements hold true across various models?**
>
>
> Thank you for your insightful suggestion! We experimented with two variants along different dimensions. The results of these experiments are presented in the tables below.
>
> First, we replaced the pretrained language model (LLM) from Vicuna1.5-7B to LLaMA3-8B, keeping all other settings unchanged. We observed that the stronger pretrained language model indeed led to improved performance on the visual CoT benchmark.
>
> Second, we explored a different model architecture for encoding visual inputs. Instead of using a 2-layer MLP projector to obtain visual tokens, we implemented a q-former (cross-attention module) to extract visual tokens from the vision encoders. The former approach is used by models like MiniGPT-4 [4], MiniGPT-v2 [1], Qwen-VL [2], and InstructBlip [5], while the latter is employed by models such as LLaVA [6], Shikra [7], Sphinx [8], and CogVLM [9]. Our experiments demonstrate that our training pipeline effectively enhances model performance across different architectures.
>
> |  Method  | DocVQA  | TextCaps  |  TextVQA  |DUDE  |  SROIE  | InfographicsVQA  | Flickr30k  |  Visual7W  |   GQA  | Open images  | VSR  | Birds-200-2011  | Average |
> |  ----  | ----  |----  | ----  |----  | ----  |----  | ----  |----  | ----  |----  | ----  | ----  | ---- |
> | LLaVA-7B (336) | 0.244 | 0.597| 0.588 | 0.290| 0.136|0.400| 0.581| 0.575| 0.534| 0.412| 0.572| 0.530 | 0.454|
> | VisCoT-7B (336)  | 0.476 | 0.675 | 0.775 | 0.386| 0.470| 0.324| 0.668| 0.558| 0.631| 0.822| 0.614 | 0.559| 0.580 |
> | VisCoT-Llama3-8B (336)  | 0.505 | 0.718 | 0.791 | 0.429| 0.494| 0.346| 0.700| 0.563| 0.668| 0.845| 0.643 | 0.578| 0.607 |
> | VisCoT-MiniGPT-v2-7B (448)  | 0.442 | 0.629 | 0.745 | 0.353| 0.451| 0.329| 0.661| 0.560| 0.612| 0.805| 0.603 | 0.548| 0.562 |
>
> **Q3: The Limitations section in the Supplementary Material didn't seem to address this: What happens if there are multiple bounding boxes needed for a proper reasoning chain, e.g., to answer "What is between [Object A] and [Object B]?"**
>
> Thank you for your insightful question. Currently, our bounding box approach is designed to identify the final region that directly answers the question. In the case of the example you mentioned, the bounding box would encompass the object located between Object A and Object B, so there would typically be a single bounding box for that object. Our current framework does not specifically localize individual objects mentioned in the question. If multiple objects are involved in answering the question, our current approach merges them into a single, larger bounding box. We appreciate your thoughtful observation, and we will consider addressing this by incorporating a multi-bounding box solution in future work.

---

> > ### Author Rebuttal · Authors · 2024-08-17
> >
> > # Continue'd...
> >
> > **Q4: Related to Q3, how should multiple bounding boxes across different entities (e.g., different pages in a document) or across different reasoning steps (e.g., "what is above the object to the right of [Object A]") be handled?**
> >
> > Thank you for your question. Just like in the scenario you mentioned, if the answer spans multiple locations, we merge them into a single bounding box. However, such cases are currently very rare. The vast majority of questions require only a single key region. Our current framework is designed to handle scenarios with a single bounding box that directly corresponds to the region ultimately resolving the question. If answering the question requires multiple regions, we currently merge these regions into a single bounding box. However, in our dataset, bounding boxes are annotated for each reasoning step (where detailed reasoning processes are annotated). Your suggestion is very insightful, and we will carefully consider it in future developments, incorporating it along with your previous suggestion.
> >
> > We hope that our response sufficiently addresses your comments. If you have any further questions or concerns, please feel free to reach out to us for further discussion.
> >
> > **Reference**
> >
> > [1] ​​Chen, Jun, et al. "Minigpt-v2: large language model as a unified interface for vision-language multi-task learning." arXiv preprint arXiv:2310.09478 (2023).
> >
> > [2] Bai, Jinze, et al. "Qwen-vl: A versatile vision-language model for understanding, localization, text reading, and beyond." (2023).
> >
> > [3] Li, Yanwei, et al. "Mini-gemini: Mining the potential of multi-modality vision language models." arXiv preprint arXiv:2403.18814 (2024).
> >
> > [4] Zhu, Deyao, et al. "Minigpt-4: Enhancing vision-language understanding with advanced large language models." arXiv preprint arXiv:2304.10592 (2023).
> >
> > [5] Dai, Wenliang et al. “InstructBLIP: Towards General-purpose Vision-Language Models with Instruction Tuning.” ArXiv abs/2305.06500 (2023): n. pag.
> >
> > [6] Liu, Haotian, et al. "Visual instruction tuning." Advances in neural information processing systems 36 (2024).
> >
> > [7] Chen, Keqin, et al. "Shikra: Unleashing multimodal llm's referential dialogue magic." arXiv preprint arXiv:2306.15195 (2023).
> >
> > [8] Lin, Ziyi, et al. "Sphinx: The joint mixing of weights, tasks, and visual embeddings for multi-modal large language models." arXiv preprint arXiv:2311.07575 (2023).
> >
> > [9] Wang, Weihan, et al. "Cogvlm: Visual expert for pretrained language models." arXiv preprint arXiv:2311.03079 (2023).

---

> > > ### Author Response · Authors · 2024-08-28
> > >
> > > Dear Reviewer 1mvY,
> > >
> > > We sincerely appreciate the time and effort you have invested in reviewing our submission. Your insightful feedback has been invaluable to us, and we have diligently worked to address all the concerns you raised in our rebuttal. As the author-reviewer discussion phase is drawing to a close, we would like to confirm whether our responses have effectively addressed your concerns. We are more than happy to provide any further details or explanations. Thank you once again for your thoughtful review and consideration.
> > >
> > > Best regards,
> > >
> > > The Authors

---

> ### Comment · Reviewer_1mvY · 2024-08-29
>
> Thanks to the authors for the additional results/comparisons, and for satisfactorily addressing the multiple bounding-box issue. I have raised my score to 7.
>
> Assuming the paper is ultimately accepted (somewhere, if not here), I would strongly suggest being more upfront/explicit about the limitations (including the multiple bounding-box issue), at least framed as future work if you prefer. No work is perfect or forever done, and I much prefer to read, cite and share papers that are frank about their limitations.
>
> Thanks.

---

> > ### Author Response · Authors · 2024-08-29
> >
> > Thank you for your feedback and for raising your score. We appreciate your suggestion to clearly outline the limitations of our work. We agree that openness about challenges and future improvements is crucial, and we will certainly emphasize these points more clearly in the final version. Thank you once again for your valuable insights and recommendations.

---

### Decision · Program_Chairs · 2024-09-26

**Decision:**

Accept (Spotlight)

**Comment:**

All reviewers have expressed their willingness to accept this paper.  Many issues were well addressed during the rebuttal stage. Therefore, I recommend accepting this work. The authors are encouraged to include necessary experimental analysis in the final version so that this work can be better understood in the future.